# Computational design of soluble and functional membrane protein analogues

Casper A. Goverde[1,7], Martin Pacesa[1,7], Nicolas Goldbach[1,7], Lars J. Dornfeld[1], Petra E. M. Balbi[1], Sandrine Georgeon[1], Stéphane Rosset[1], Srajan Kapoor[2], Jagrity Choudhury[2], Justas Dauparas[3,4], Christian Schellhaas[1], Simon Kozlov[5], David Baker[3,4,6], Sergey Ovchinnikov[5], Alex J. Vecchio[2] & Bruno E. Correia[1✉]

De novo design of complex protein folds using solely computational means remains a substantial challenge[1]. Here we use a robust deep learning pipeline to design complex folds and soluble analogues of integral membrane proteins. Unique membrane topologies, such as those from G-protein-coupled receptors[2], are not found in the soluble proteome, and we demonstrate that their structural features can be recapitulated in solution. Biophysical analyses demonstrate the high thermal stability of the designs, and experimental structures show remarkable design accuracy. The soluble analogues were functionalized with native structural motifs, as a proof of concept for bringing membrane protein functions to the soluble proteome, potentially enabling new approaches in drug discovery. In summary, we have designed complex protein topologies and enriched them with functionalities from membrane proteins, with high experimental success rates, leading to a de facto expansion of the functional soluble fold space.

Protein design enables the expansion of nature's molecular machinery, creating synthetic proteins with new functionalities. Traditionally, protein design has been dominated by physics-based approaches, such as Rosetta[3]. However, these methods require parametric and symmetric restraints to guide the design process and often extensive experimental screening and optimization. This proves problematic for the design of functional proteins with complex structural topologies. Recently, structure prediction pipelines, such as AlphaFold2 (AF2)[4], have achieved unprecedented accuracy in predicting protein structure given the amino acid sequence. With the rise of deep learning-based methods, exploring the sequence space has become increasingly feasible, allowing the discovery of proteins with stable topologies and new functions. Deep learning-powered methods have also been influential in various tasks that include the generation of new designable backbones[5–7], oligomeric protein assemblies[8,9], proteins with embedded functional motifs[10], new protein structural descriptors[11], the sequence design problem[12,13] and, more recently, the generation of a diverse range of protein topologies using diffusion models[9,14,15]. In addition, structure prediction networks can be inverted and used for protein design, resulting in the generation of plausible protein backbones[6,7,16].

Nevertheless, designing protein folds with complex structures, including non-local topologies and large sizes, remains challenging; however, it is essential for creating new protein functions. In addition to design proficiency, the answers to many questions about the fundamental determinants of protein structure and folding remain elusive, particularly regarding the generalizability of deep learning methods beyond natural protein structures and sequences. To probe some of these questions, we analysed the protein fold space in the Structural Classification of Proteins (SCOP) database[17] and observed a segregation at the structural level between proteins in the soluble proteome and those in the cell membrane environment (Fig. 1a). We observed that 1,075 membrane proteins exhibited unique topologies that were not found in soluble form, with only 189 folds being present in both soluble and membrane environments. This raises the question of whether integral membrane protein topologies have some fundamental structural features that preclude them from existing in the soluble fold space. Consequently, we investigated whether membrane folds could be designed as soluble analogues, thus achieving a de facto fold expansion of the soluble proteome and creating opportunities for designing new functions using these previously inaccessible protein folds. Although there has been previous work on the solubilization of near-native membrane proteins using physics-based and empirical methods[18–21], no generalizable approach for the computational design of soluble membrane topologies with preserved functional aspects has been devised.

To this end, we developed a computational pipeline for robust de novo protein design, based on inversion of the AF2 network[7] coupled with sequence design using ProteinMPNN[13] (Fig. 1b). Our approach allowed us to computationally design highly stable folds that were previously very challenging (Ig-like fold (IGF), β-barrel (BBF) and TIM-barrel (TBF)), as well as soluble analogues of integral membrane protein folds (claudin, rhomboid protease, G-protein-coupled receptor (GPCR)) without the need for parametric design restraints or subsequent experimental optimization. Finally, we demonstrated that the soluble analogues could be designed in a conformation-specific manner while preserving native functional motifs with structurally

[1]Laboratory of Protein Design and Immunoengineering, Ecole Polytechnique Fédérale de Lausanne and Swiss Institute of Bioinformatics, Lausanne, Switzerland. [2]Department of Structural Biology, University at Buffalo, Buffalo, NY, USA. [3]Department of Biochemistry, University of Washington, Seattle, WA, USA. [4]Institute for Protein Design, University of Washington, Seattle, WA, USA. [5]Department of Biology, Massachusetts Institute of Technology, Cambridge, MA, USA. [6]Howard Hughes Medical Institute, University of Washington, Seattle, WA, USA. [7]These authors contributed equally: Casper A. Goverde, Martin Pacesa, Nicolas Goldbach. ✉e-mail: bruno.correia@epfl.ch

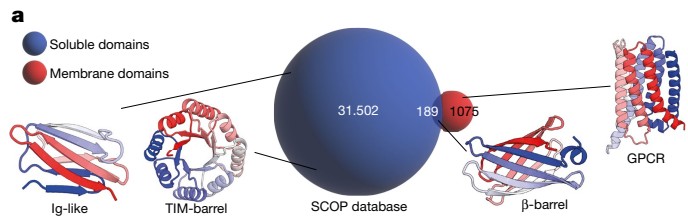

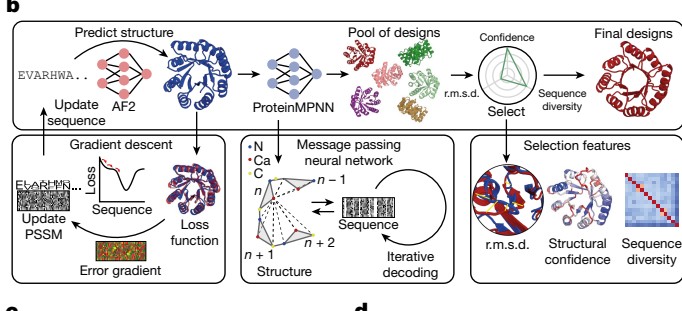

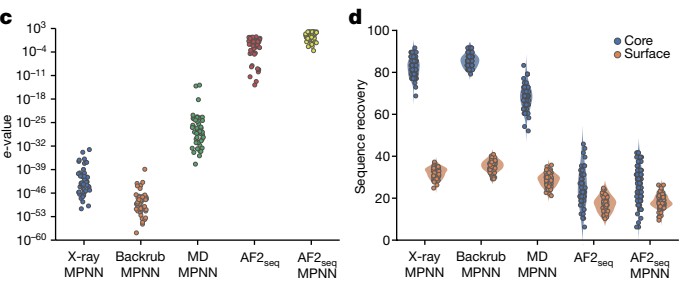

**Fig. 1 | Overview of the fold space across different environments and computational design approach. a**, Overview of the occurrence of soluble and membrane folds in the SCOP structural database, with depictions of selected representatives. **b**, Schematic representation of the integrated design pipeline for backbone and sequence generation. Given a target structure, an initial sequence is generated using AF2 through loss function optimization. The resulting structure is then passed to ProteinMPNN to sample new amino acid sequences for a given fold. ProteinMPNN designs are filtered on the basis of structural similarity to the target, confidence and sequence diversity. **c**, Novelty of generated sequences resulting from different backbone sampling methods, evaluated by $e$-values relative to the non-redundant protein sequence database. **d**, Sequence recovery of core and surface residues of TBF ProteinMPNN designs generated on the basis of the reference crystal X-ray structure (Protein Data Bank (PDB) 5BVL), Rosetta-perturbed backbones (backrub protocol), molecular dynamics simulation trajectories or AF2$_{seq}$-generated structures.

elaborate features and of biological and therapeutic relevance, such as G-protein-binding interfaces and toxin–receptor interaction sites. Our findings showcase the remarkable success and accuracy of deep learning-based methods in protein design, paving the way for exploration of new protein topologies and sequences for improved functional design strategies.

## Structure–sequence generation using deep learning

AF2-based design approaches have been shown to generate plausible protein backbones[7,22,23]; however, their performance in sequence design has been suboptimal, as evidenced by low experimental success rates[7,8,13]. Wicky and coworkers[8] have demonstrated the efficiency of using ProteinMPNN on AF2-generated structures to enhance their expression and solubility, but it remained unclear whether this approach could be successfully used to explore the sequence space of complex protein folds with intricate topological features, including those only found in membrane environments (Fig. 1a). To address this challenge, we integrated our previously developed

AF2-based design approach (AF2$_{seq}$)[7] with the ProteinMPNN framework (Fig. 1b).

In this pipeline, we use AF2$_{seq}$ to generate sequences that adopt a desired target fold. AF2$_{seq}$ optimizes a sequence on the basis of a loss function that comprises both topological and structural confidence loss components (Methods) until a sequence is found that folds to the desired topology. We then apply ProteinMPNN sequence optimization to the AF2$_{seq}$-generated starting topologies. Finally, the structures of all resulting sequences are repredicted with AF2 and filtered on the basis of their structural similarity to the target topology (template modelling (TM) score > 0.8), confidence scores (predicted value of the local distance difference test (pLDDT) > 80) and sequence novelty relative to naturally occurring sequences ($e$-value > 0.1).

In silico assessment showed that despite the restricted structural diversity (Extended Data Fig. 1a,b), AF2$_{seq}$-designed backbones enabled ProteinMPNN to generate much greater protein sequence diversity for a desired fold than that of classical backbone sampling methods such as Rosetta Backrub[24] or molecular dynamics simulations (Fig. 1c and Extended Data Fig. 1a,b). To investigate the source of the diversity, we examined sequence conservation at the core and surface of the designs following ProteinMPNN optimization, which was originally reported to consistently recover approximately 50% of the starting sequence[13]. Sequence optimization using ProteinMPNN alone resulted in high sequence recoveries in the core of the designs, relative to the starting sequence (Fig. 1d). AF2$_{seq}$-generated designs exhibited low sequence recoveries in both the core and the surface compared with the sequence of the target protein. This indicates that the novelty and designability of our backbones primarily stem from the new backbone templates generated by AF2$_{seq}$. Increasing levels of Gaussian noise applied to the backbone before ProteinMPNN sequence design could also reduce sequence recovery (Extended Data Fig. 1c); however, this was at the expense of low-confidence predictions that deviated significantly from the target fold (Extended Data Fig. 1d–f). In addition, we found that for some more complex design tasks, the target structure could not be predicted in single sequence mode by AF2 after ProteinMPNN redesign. However, when using a combination of AF2$_{seq}$ and ProteinMPNN (AF2$_{seq}$-MPNN), we found the input sequence to result in accurate structural predictions of the target folds (Extended Data Fig. 2a–d). Therefore, we sought to test whether our design strategy would be successful in designing protein folds that have thus far been challenging to other approaches.

## Design of topologically complex folds

To identify challenging design targets for our pipeline, we quantified the topological complexity of protein folds using metrics of protein length and sequence contact order (Extended Data Fig. 3a,b and Methods). On the basis of this assessment, and given how challenging some folds have been for computational design, we selected three folds to test our approach: the IGF, BBF and TBF (Fig. 2a). The IGF is one of the most prevalent folds in nature and is an essential building block of immunological effectors and therapeutics such as antibodies and receptors[25]. The IGF consists of two stacked β-sheets, presenting a substantial design challenge. This is because of its non-local interactions and susceptibility to aggregation through edge β-strands[26], previously requiring strict parametric and symmetry restraints during design[27,28]. Using our AF2$_{seq}$-MPNN protocol (Fig. 1b), we designed IGFs that were significantly distant from natural protein sequences (Fig. 2b). We selected 19 designs for experimental characterization on the basis of AF2 confidence scores and sequence diversity (Supplementary Figs. 1 and 2). Seven designs were soluble, with four designs exhibiting monodisperse peaks in solution (Supplementary Fig. 3). Exemplified by IGF_10 (Fig. 2c), the designed IGFs exhibited a typical β-sheet-rich secondary structure profile according to circular dichroism spectroscopy, together with unusually high thermostability[29] (Fig. 2c and Supplementary Fig. 3).

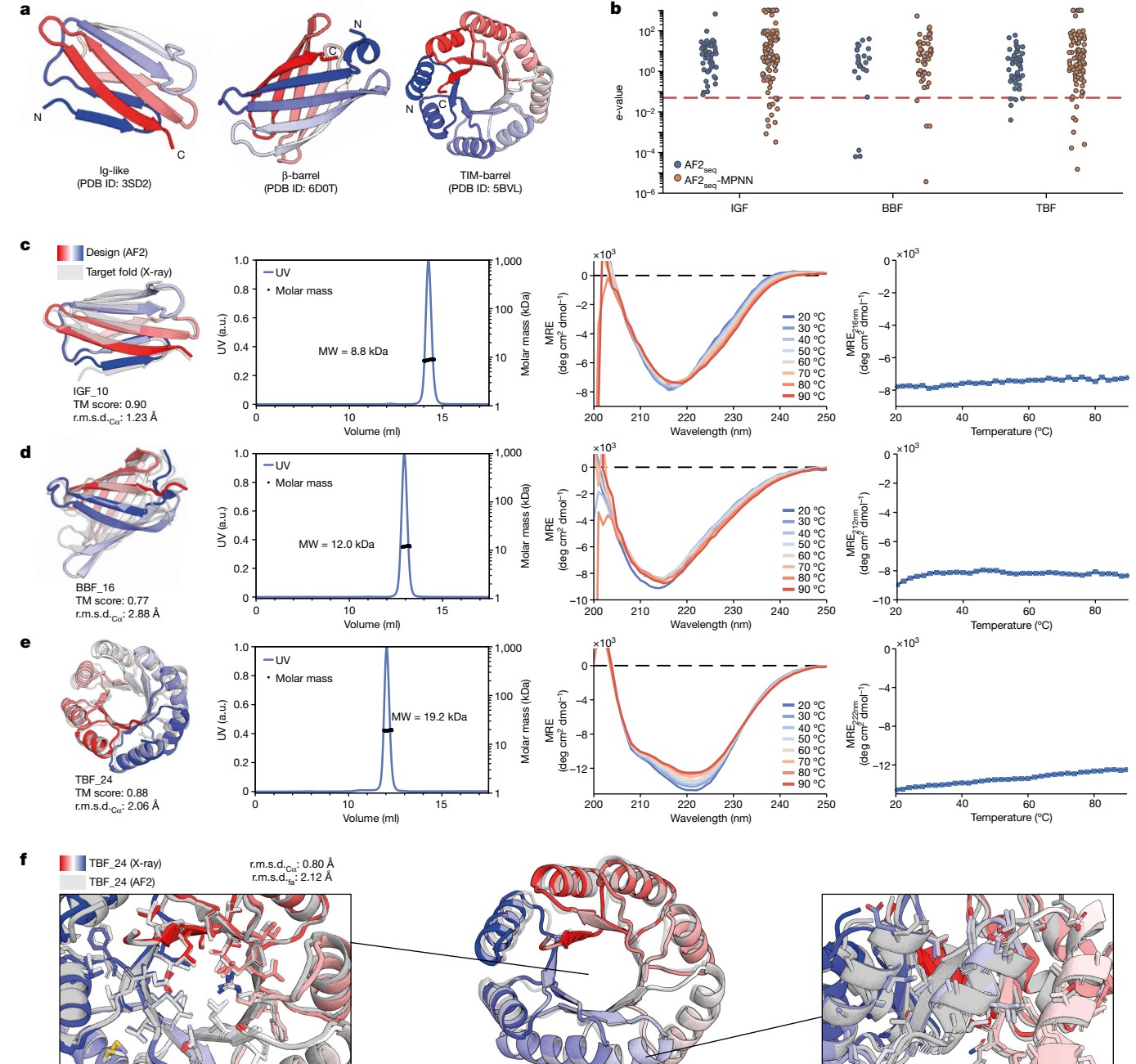

**Fig. 2 | Experimental characterization of designed complex protein topologies. a**, Cartoon depiction of three protein topologies that have been challenging for computational design: IGF, BBF and TBF. **b**, Closest *e*-value hits for the AF2$_{seq}$ and AF2$_{seq}$-MPNN designs when searching a non-redundant protein sequence database. The significance threshold of 0.05 is highlighted, indicating little sequence homology with natural sequences. **c**–**e**, Characterization of designs IGF_10 (**c**), BBF_16 (**d**) and TBF_24 (**e**) showing superposition of the design (colour) and the target fold (grey), the corresponding SEC–MALS measurement, circular dichroism spectra at different incubation temperatures and the circular dichroism melting curve. **f**, X-ray structure of TBF_24 (coloured) superimposed on the design model (grey).

Next, we attempted to design a de novo BBF, a fold present both in the soluble and membrane proteomes, with applications in small-molecule binders, transporters and sensors[30–33]. It consists of eight antiparallel β-strands with precise hydrogen bonding patterns[30], making its design extremely challenging. Previously, Dou et al. used a set of design principles that involve the introduction of glycine kinks, β-bulges and tryptophan corners to alleviate backbone strain and allow continuous hydrogen bonding connectivity[30]. We investigated whether our approach could be used to successfully design BBFs without explicitly defining such constraints. We experimentally characterized 25 designs,

of which six were found to be folded and monomeric in solution while exhibiting high thermal stability (Fig. 2d and Supplementary Fig. 4). Sequence analysis of the designs showed high glycine residue recovery at glycine kink positions (Extended Data Fig. 4a–c), as observed by Verkuil et al.[34]. This demonstrates that not all empirically derived features are necessary for successful BBF design, and that a larger uncharted sequence space can be explored.

Finally, we attempted to design a TBF, a challenging protein topology that is of paramount importance in biology, as its structure is highly proficient in supporting enzymatic active sites, making it an ideal

candidate for the design of enzymes with new catalytic functions[35]. The TBF comprises eight parallel-paired β-strands, each separated by an α-helix, resulting in long-range interactions between the β-strands[36]. The TBF has been a longstanding challenge in protein design[36,37], and it is only very recently that several studies have successfully designed this fold[9,38,39]. Previous TBF design strategies imposed symmetry and parametric restraints at both the structural and sequence levels[38,39]. With our pipeline, we could design TBFs without any constraints, allowing greater structural and sequence diversity and even asymmetry, which could potentially accommodate more complex enzymatic sites (Supplementary Fig. 5). We experimentally assessed 25 designs, five of which were monomeric, folded and highly thermostable in solution (Fig. 2e and Supplementary Fig. 6). To confirm the accuracy of our design, we solved a crystal structure of TBF_24 at 1.34 Å resolution (Fig. 2f). Our asymmetric design showed noticeable structural deviations from the initial symmetric template (Fig. 2f), with an overall backbone r.m.s.d._Cα (root mean square deviation computed over the Cα atoms of the backbone) of 2.06 Å (Extended Data Fig. 4d). Comparison of the X-ray structure with the designed model showed the r.m.s.d._Cα and full-atom r.m.s.d._fa (root mean square deviation computed over all the atoms in the structure) were 0.80 Å and 2.12 Å, respectively (Fig. 2f). These structural comparisons demonstrate the remarkable accuracy of our design approach, further underlined by the almost identical side-chain placement in both the core and peripheral regions of the protein (Fig. 2f). Given the encouraging results obtained with our initial designs, we considered whether our approach would allow us to probe the sequence space of topologies not present in the soluble proteome, such as those of integral membrane proteins.

## Solubilizing the membrane protein fold space

In a domain analysis performed over the SCOP database, we observed that both the soluble and membrane proteome each encompassed a group of unique structural protein topologies, with only a narrow overlap between the two (Figs. 1a and 3a). This prompted us to ask whether it was possible to design soluble analogues of such membrane-only folds or whether they contained intrinsic structural features that precluded them from existing in soluble form. Previous studies have demonstrated that simply substituting exposed hydrophobic residues for polar or charged amino acids might not be sufficient to solubilize these folds, as the interactions between the surface residues have to be carefully considered[18,40–42]. To address this question, we set out to design soluble analogues of membrane proteins using the AF2_seq-MPNN pipeline (Methods). We selected three membrane folds to test the design strategy: the claudin fold[43], the rhomboid protease fold[44] and the GPCR fold[2] (Fig. 3b).

Initial designs by AF2_seq exhibited high sequence novelty compared with natural proteins (Fig. 3c) and a low fraction of surface hydrophobics (Fig. 3d); however, none could be expressed in soluble form (Fig. 3e). We attempted to optimize the sequences using the standard ProteinMPNN model, but the resulting sequences consistently recovered the surface hydrophobics, probably owing to the similarity of the topology to that of membrane proteins encountered during training (Fig. 3d). Biasing the amino acid sampling towards hydrophilic amino acids (AF2_seq-MPNN_bias) only marginally improved the solubility of the designs (Fig. 3d). Therefore, we retrained the ProteinMPNN network using a dataset of only soluble proteins, which we named soluble MPNN (MPNN_sol) (Fig. 3c–e). AF2_seq-MPNN_sol was able to produce new sequences with high confidence scores predicted by AF2 and low fraction of surface hydrophobics (Fig. 3d and Supplementary Fig. 1f). As a result, we were able to generate high-confidence designs of membrane protein topologies that do not exist in the soluble proteome.

We started by designing soluble analogues of the claudin fold, a class of proteins involved in the formation of tight junctions, which are critical in controlling the flow of molecules between layers of epithelial and endothelial cells[43]. Claudin folds are composed of an α/β mixed secondary structure in which there are four transmembrane α-helices and an extracellular β-sheet[43]. The composition of the β-sheet determines the type of tight junction between cells that is being formed, resulting in highly selective permeability of ions and solutes[43]. Claudin-targeting therapies hold great promise as new cancer therapies, and soluble claudin analogues could represent a new route to screen for claudin binders[45]. We tested 13 designs for the claudin-like fold (CLF), of which ten were found to be expressed in soluble form (Fig. 3e). Five designs were further biochemically characterized; three were monomers in solution according to size-exclusion chromatography with multi-angle light scattering (SEC–MALS) and were folded, with two showing a melting temperature ($T_m$) above 90 °C (Fig. 3f and Supplementary Fig. 7). The CLF designs showed sequence identity below 13% relative to the native fold and nearest $e$-values to natural sequences below 0.063 (Fig. 3c and Supplementary Fig. 1). AF2-predicted structures, comparison with the designed models, exhibited r.m.s.d._Cα values ranging from 2.84 to 4.03 Å (Supplementary Fig. 7). The CLF design series showed that our approach could be used successfully to design soluble analogues of membrane proteins with simple membrane-spanning topologies such as four-helix bundles.

Next, we attempted to design a larger fold and a more intricate topology, the rhomboid protease fold (RPF). The RPF comprises six transmembrane α-helices, with many structured loops and long-range contacts[46] (Fig. 3b). In addition, it harbours a serine–histidine catalytic dyad buried in the cell membrane, allowing it to cleave transmembrane protein domains and play an important part in cell signalling, which makes it a therapeutically interesting target[46]. We selected 15 designs for protein expression, of which 13 were found to be soluble (Fig. 3e) and five were selected for further experimental characterization. Three of the five designs showed a single monomeric species in solution and the expected helical secondary structure as assessed by circular dichroism (Fig. 3g and Supplementary Fig. 8). All of the three monomeric species of RPFs exhibited high thermal stability, with $T_m$ above 90 °C (Fig. 3g and Supplementary Fig. 8). Notably, the AF2 structure predictions for the designs were less accurate than those for the CLF designs, with the r.m.s.d._Cα values between models and predictions ranging from 3.34 to 5.57 Å (Fig. 3g and Supplementary Fig. 8). Overall, the high r.m.s.d._Cα values between design models and AF2 predictions highlight the inherent difficulty in designing folds with such structural complexity.

Then, we tested our design approach in one of the most prevalent membrane folds in nature: the GPCR fold. GPCRs are the largest and most diverse family of membrane receptors in eukaryotes, playing important parts in signalling pathways[2]. About 34% of all drugs approved by the US Food and Drug Administration target GPCRs, and they remain the most studied drug target[47]. The core topology of GPCRs comprises seven transmembrane helices that facilitate numerous non-local interactions, enabling them to bind to a variety of ligands, including photoreceptors, odours, pheromones, hormones and neurotransmitters[2]. De novo design of GPCR-like folds (GLFs) offers the potential to create new small-molecule receptors and protein scaffolds with functional sites of GPCRs. We tested 56 designs, of which 36 were expressed to be soluble, and we selected the ten most highly expressed designs for further biochemical characterization. Of these ten designs, nine were monodisperse monomers in solution, and all showed the characteristic circular dichroism signature of α-helix-rich proteins (Fig. 3h and Supplementary Fig. 9). All ten designs also showed high thermal stabilities ($T_m$ > 90 °C) (Fig. 3h and Supplementary Fig. 9).

To assess the design accuracy, we attempted to crystallize the soluble analogues of membrane topologies and obtained high-resolution X-ray structures for one claudin, one rhomboid protease and two GPCR designed folds (Fig. 4 and Extended Data Fig. 5a). The CLF_4 structure exhibited exceptional design precision in both backbone and side chains, as indicated by an r.m.s.d._Cα of 0.73 Å and r.m.s.d._fa

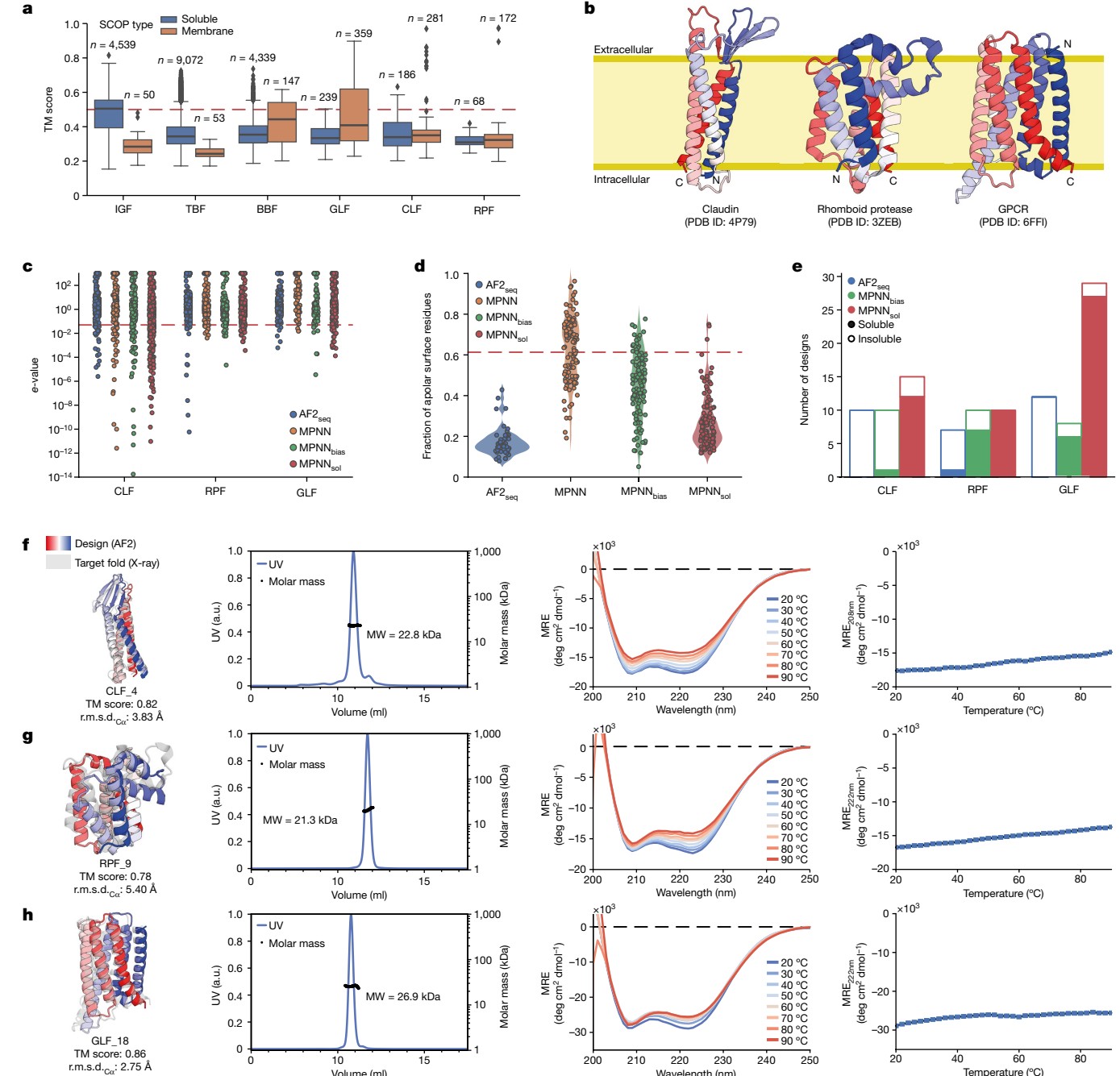

**Fig. 3 | Experimental characterization of soluble analogues of membrane proteins. a**, Structural similarity for each of the target folds against the SCOP database. TM score cut-off of 0.5 is highlighted, denoting significant structural similarity to the reference fold. The centre line represents the median of the data (50th percentile), whereas the box (coloured) represents the 25th and 75th percentiles of the values. The whiskers show the minimum and maximum values of the distribution. Data points were considered to be outliers (black diamonds) if they fell outside the 1.5 interquartile range. **b**, Cartoon representation of three transmembrane topologies chosen to be redesigned as soluble folds: the CLF, RPF and GLF. **c**, Closest *e*-value hits of the solubilized CLF, RPF and GLF against a non-redundant protein sequence database. Most of the designed

sequences differed substantially from natural sequences, as indicated by *e*-values higher than the significance threshold of 0.05 (red line). **d**, Fraction of hydrophobic residues found on the surface of the GLF designs using different sequence-generation methods following AF2$_{seq}$ backbone generation. The fraction of surface hydrophobics of the native GLF is 0.61 (red line). **e**, Number of designs resulting in soluble expression of the designed soluble membrane protein analogues. **f**–**h**, Experimental characterization of CLF_4 (**f**), RPF_9 (**g**) and GLF_18 (**h**). Comparison of the design (colour) and target fold (grey) solution behaviour by SEC–MALS, circular dichroism spectra at different incubation temperatures and melting temperature profiles by circular dichroism.

of 1.28 Å (Fig. 4a). Comparison between CLF_4 and the native claudin demonstrated accurate secondary structural element positioning, with an r.m.s.d.$_{C\alpha}$ of 3.63 Å and most of the deviation arising from the β-sheet region (Fig. 4b and Extended Data Fig. 6a,b). In addition, the four helices were mostly hydrophilic, as evidenced by the low

lipophilicity potential on the surface (Fig. 4c). In the case of the rhomboid protease design RPF_9, we observed high accuracy between the X-ray structure and the design model, with r.m.s.d.$_{C\alpha}$ of 0.97 Å and r.m.s.d.$_{rfa}$ of 1.83 Å. However, the structural similarity was significantly lower compared with the target fold, as indicated by an r.m.s.d.$_{C\alpha}$ of

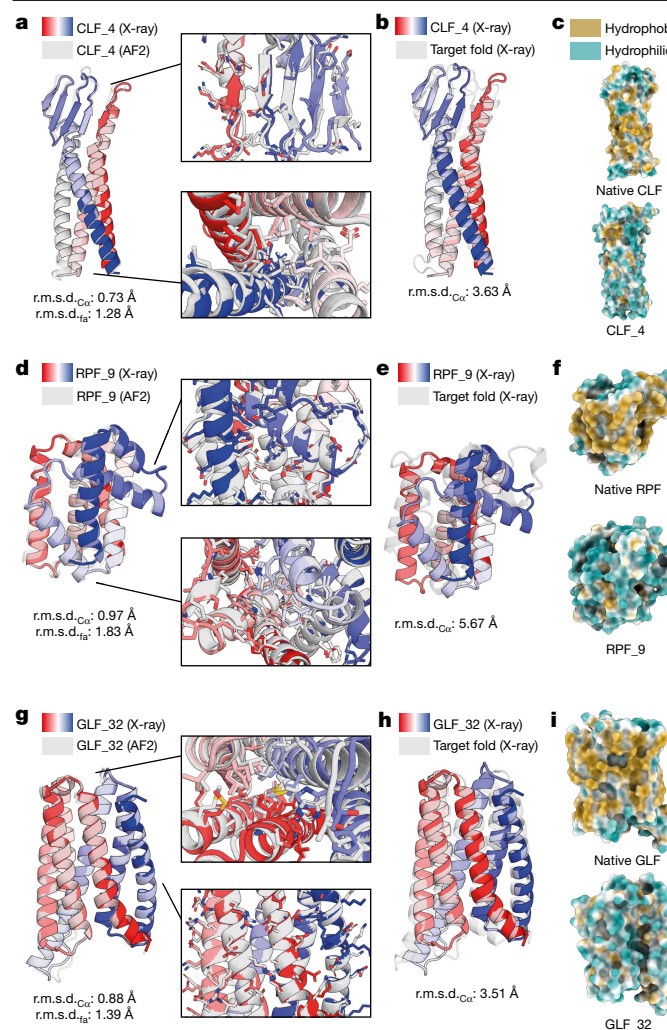

**Fig. 4 | Soluble analogues of membrane proteins solved by X-ray crystallography. a**, X-ray structure of CLF_4 (coloured) superimposed on the design model (grey). **b**, X-ray structure of CLF_4 (coloured) superimposed on the design model (grey). **c**, Molecular lipophilicity potential of the surface of the claudin design target and the soluble design CLF_4. **d**, X-ray structure of RPF_9 (coloured) superimposed on the design model (grey). **e**, X-ray structure of RPF_9 (coloured) superimposed on the design model (grey). **f**, Molecular lipophilicity potential of the surface of the rhomboid protease design target and the soluble design RPF_9. **g**, X-ray structure of GLF_32 (coloured) superimposed on the design model (grey). **h**, X-ray structure of GLF_32 (coloured) superimposed on the design model (grey). **i**, Molecular lipophilicity potential of the surface of the GPCR design target and the soluble GLF_32 design. After redesign of the original membrane folds with MPNN$_{sol}$, the hydrophobicity (yellow) of the surface was significantly reduced, and polarity was increased (blue).

5.67 Å (Fig. 4d–e). Specifically, large structural deviations in the first extracellular loop (Extended Data Fig. 6c,d) were observed, which could have been due to its native positioning in the water–membrane interface[46]. The designed RPF_9 fold showed significantly increased hydrophobicity on the transmembrane surface compared with the native RPF fold (Fig. 4f). Last, the designed GLFs preserved the canonical seven-helical bundle characteristic of native GPCRs. Structurally, the crystal structures were in very good agreement with the design models, with r.m.s.d.$_{C\alpha}$ values of 1.05 Å and 0.88 Å for GLF_18 and GLF_32, respectively (Fig. 4g,h and Extended Data Fig. 5a). This accuracy further extended to the side-chain level, for which comparisons of crystal structures versus design models for GLF_18 and GLF_32 showed 1.54 and 1.40 Å r.m.s.d.$_{fa}$ values, respectively. Comparing the structures

of the soluble analogues with that of the reference native GPCR, the overall backbone r.m.s.d.$_{C\alpha}$ values were 3.08 and 3.51 Å for GLF_18 and GLF_32, respectively (Fig. 4g,h and Extended Data Fig. 5b). By analysing the lipophilicity potential at the surface[48] of the designed proteins, we observed a clear transition from an initially hydrophobic surface to a hydrophilic one (Fig. 4i and Extended Data Fig. 5c). Interestingly, many of the sequence signatures of the GPCR fold were absent from our designs, including the evolutionarily conserved DRY motif in the first intracellular loop[49], the (N/D)PxxY motif in the seventh helix[50] and the transmembrane proline-rich domains[51] (Extended Data Fig. 7a–c). This demonstrates that by design one can explore very diverse sequence spaces while removing potential evolutionary biases. At the structural level, we observed that the irregular local structure of the terminal helix was preserved, whereas the intracellular segment of the GPCR fold exhibited notable structural deviation in GLF_18 (Extended Data Fig. 7d–g). Our results demonstrate that integral membrane folds can be successfully designed in soluble form, hinting that these topologies share many of the designability principles and constraints of folds present in the soluble proteome.

## Functional soluble membrane protein analogues

After validating the structural accuracy of our designs, we explored the possibility of functionalizing the designed soluble analogues. To this end, we devised an approach in which we explicitly fix structural segments and amino acid identities of the functional motifs during design, whereas the transmembrane segment is solubilized in their context (Fig. 5). We applied this strategy to the design of soluble analogues of human claudin-1 and claudin-4 (ref. 52), in which varying levels of the two extracellular segments were preserved (Methods). To verify structure and function, we tested their binding to *Clostridium perfringens* enterotoxin (CpE), a common foodborne pathogen to humans known to bind claudin-1 and claudin-4 differentially[52]. Binding assays using bio-layer interferometry (BLI) indicated that soluble claudin-1 and claudin-4 exhibit binding kinetics and affinities for CpE that are comparable with those of their membrane-bound counterparts[52] (Fig. 5b–e). The claudin-1 designs exhibited lower binding affinity for CpE versus claudin-4, owing to the latter being a high-affinity CpE receptor. Notably, the higher proportion of native sequence preserved in claudin-1 design CLN1_14 resulted in a reduced melting temperature compared with CLN1_18 (Extended Data Fig. 8a–d).

Additionally, we observed that CLN4_20 assembled into soluble high-molecular-weight oligomeric species, according to SEC–MALS (Fig. 5f,g). The oligomeric assemblies could be disrupted by addition of the carboxy-terminal claudin-binding domain of CpE (cCpE) (Fig. 5g–i). This is analogous to the disassembly of high-order claudin oligomers within tight junctions by cCpE in the gut[52,53]. To confirm that the soluble analogue engaged the toxin in the same manner as the natural membrane-bound claudin-4, we reconstituted the complex together with a Fab and nanobody to increase its size and determined its structure using cryo-electron microscopy (cryo-EM) (Fig. 5j,k and Extended Data Fig. 8e). We observed that both the claudin topology and the toxin binding mode were comparable with those of the natural complex[52]. These results indicate that the designed soluble membrane protein analogues might accommodate natural sequences and functional motifs in native-like conformations, and that certain mechanistic aspects could be recapitulated in solution.

To embed function in the soluble GPCR analogues, we used two distinct design approaches. First, we created chimeric proteins from GLFs and intracellular loop 3 (ICL3) of the ghrelin receptor[54] (Fig. 6a). The residues grafted from ICL3 connect TM5 and TM6 and form hydrophobic interactions with the α subunit of G$_i$ in the activated ghrelin receptor[55]. The GLF–ghrelin chimeras were generated using a sequence transplant of the natural epitope into the corresponding region of

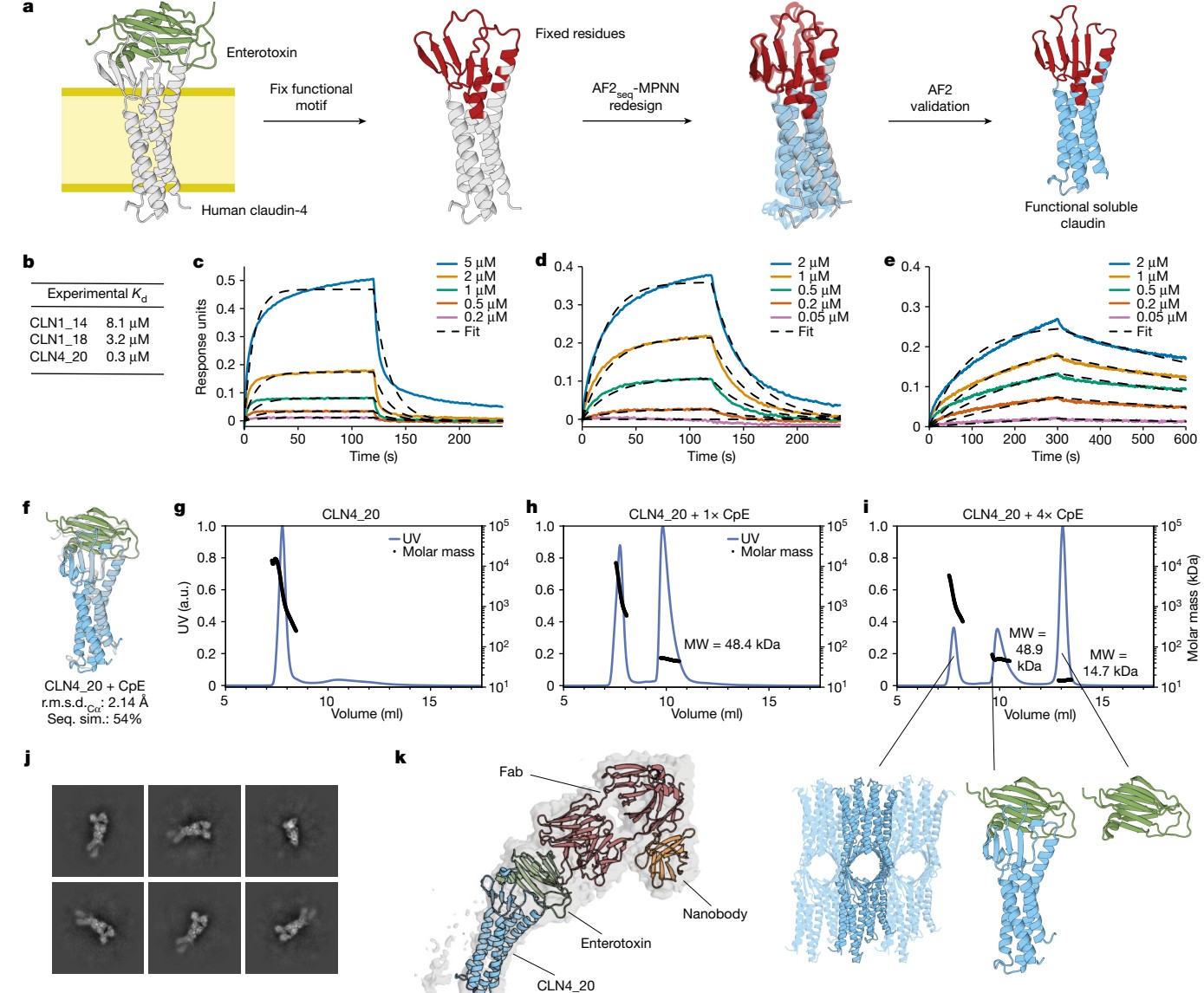

**Fig. 5 | Functionalization of soluble analogues of claudin proteins. a**, Design workflow for solubilizing claudins with fixed functional residues. CpE is known to bind to human claudin-1 and claudin-4. **b**, Binding affinities derived from kinetic measurements for binding of solubilized claudins to CpE. **c–e**, Binding kinetics for binding of solubilized claudins CLN1_14 (**c**) CLN1_18 (**d**) and CLN4_20 (**e**) to CpE. Association and dissociation during BLI are shown as solid lines and the respective fits as dashed lines. **f**, Cartoon depiction of design model of CLN4_20 bound to cCpE toxin (coloured) overlaid with the target fold (grey). **g–i**, SEC–MALS analysis of CLN4_20 mixed with 0× (**g**), 1× (**h**) or 4× (**i**) molar excess of CpE toxin. **j**, Representative two-dimensional classes of CLN4_20 bound to cCpE toxin, COP2 Fab and a nanobody. **k**, Model of CLN4_20 complex docked into reconstructed cryo-EM density.

the GLF scaffold (Fig. 6a). Using a pull-down assay, we prescreened 16 chimeric designs for binding and found that nine designs bound to the ICL3-targeting antibody[54], whereas GLF scaffolds without the ICL3 did not show any binding (Supplementary Fig. 10). We measured the binding affinity of five designs using surface plasmon resonance (SPR) and obtained affinity constant ($K_d$) values between 150 nM and 790 nM (Fig. 6b–d), whereas knockout mutants and GLF scaffolds without the epitope did not exhibit binding to the ghrelin receptor ICL3-specific antibody (Fig. 6c,d).

An important part of GPCR receptor function is the activation of intracellular signalling pathways mediated by G protein binding[2]. To recapitulate this functional aspect, we designed soluble analogues of the adenosine A2A receptor in a conformation-specific manner. This entailed the preservation of the G-protein-binding site, including evolutionarily conserved sequences, such as the DRY motif that is essential for receptor activation and G protein binding[49]. This resulted

in designs in both the active[56] and inactive[57] states with identical fixed G-protein-interacting residues (Fig. 6e and Methods). The active state is characterized by a shift of transmembrane helix 6 (TM6) in an outwards rotational motion, exposing the G-protein-binding site[58]. We characterized three constitutively active (aGLF) and three constitutively inactive (iGLF) soluble GPCR analogues; all were found to be monomeric and folded in solution (Extended Data Fig. 9a–c). SPR binding experiments with mini-$G_s$-414 (ref. 59) showed no binding to the iGLFs (Fig. 6f and Extended Data Fig. 9d), whereas a clear binding signal was observed for the aGLF designs (Fig. 6g and Extended Data Fig. 10a–d); however, exact affinities could not be determined owing to rapid interaction kinetics. To validate the specificity of the binding mode, we mutated the highly conserved DRY motif and observed that binding to the aGLFs was completely abolished (Fig. 6h and Extended Data Fig. 10d). Mutation of residues in the G-protein-binding site, outside the DRY motif, were also found to diminish or impair binding of

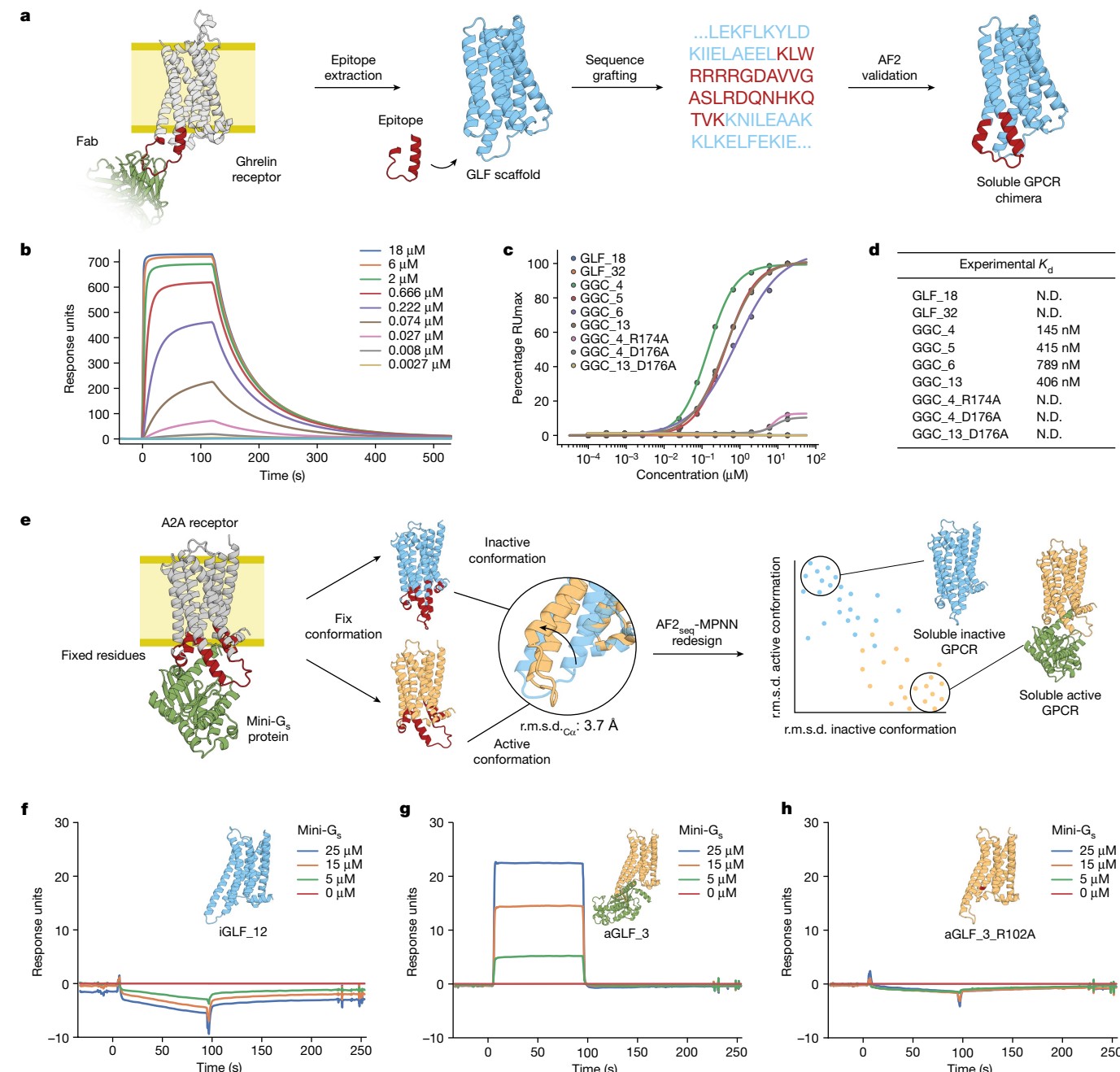

**Fig. 6 | Functionalization of soluble analogues of GPCR proteins. a**, Design of a workflow for functionalization of soluble scaffolds through grafting of the native epitope corresponding to the ICL3 loop of the ghrelin GPCR receptor that can be probed with a Fab[54] (PDB 6KO5). **b**, Representative SPR sensorgram displaying binding kinetics of increasing concentrations of ghrelin targeting antibody binding to GLF–ghrelin chimera 4 (GGC_4). **c**, Binding affinities determined by SPR of the designed GGC constructs and corresponding negative controls. **d**, Table summarizing experimental affinity constants from data shown in **c**. N.D. indicates that $K_d$ values could not be extrapolated confidently. **e**, Design of a workflow for conformation-specific design of the active[56] (PDB 5G53) and inactive[57] (PDB 3VGA) forms of the adenosine A2A receptor to facilitate or preclude mini-$G_s$ protein binding. **f**–**h**, SPR sensorgram of the inactive form iGLF_12 (**f**), active form aGLF_3 (**g**) and binding knockout mutant of aGLF_3 soluble analogue (**h**).

mini-$G_s$-414 to aGLFs (Extended Data Fig. 10d). These results indicate that specific functional states can be designed with high accuracy while preserving critical evolutionarily conserved motifs in de novo designed scaffolds.

In summary, we present here a computational approach enabling conformation-specific design and functionalization of soluble membrane protein analogues through motif grafting and constrained design procedures, which could have a range of important applications in the computational design of functional proteins and accelerated discovery of novel therapeutics.

## Conclusions

The robust computational design of complex protein folds remains a difficult endeavour. Here we present a computational approach based on deep learning that enables efficient search of non-natural sequences for a variety of protein topologies through generation of high-quality protein backbones. The computational framework based on AF2$_{seq}$-MPNN is flexible and generalizable, avoiding the need to perform fold-specific retraining or provide tedious parametric and symmetric design restraints for fold conditioning. We designed and

characterized several folds that have been very challenging to engineer with previous methods, achieving high experimental success rates in terms of soluble and folded designs. Structural characterization of the designs showed that the computational models had very high accuracy, in terms of both overall fold and the fine details of the side-chain conformations, which are critical for the design of function. In addition, we aimed to test the ability of the computational approach to expand the soluble fold space and enable the design of analogues of protein topologies only found in membrane environments. By allowing full sequence design, we designed three different membrane fold analogues, including two with highly elaborate helical topologies (rhomboid protease and GPCR) and showed that such designs were folded and monomeric in solution. The experimental structures showed once again that the design method was very accurate, and that we had recreated soluble analogues for both the rhomboid protease fold and the canonical seven-helix GPCR fold, which are not present in the soluble fold space. By doing so, we showed that membrane protein folds generally follow the same design principles as soluble protein folds, and that many such folds can be readily designed in soluble form.

Moreover, we propose that this could promote the designability of functional proteins by enabling access to a plethora of folds that are not present in the soluble fold space. Another exciting perspective is the creation of soluble analogues of membrane proteins that retain many of the native features of the original membrane proteins, such as enzymatic or ligand-binding functions; this could greatly accelerate the study of the function of these proteins in more biochemically accessible soluble formats. We demonstrated the potential of our method by incorporating native structural motifs into designed soluble analogues. By designing soluble analogues in the context of the natural functional site, we preserved even complex structural features of the sites, such as the extracellular β-sheeted domains of claudins. Recent studies have identified claudins as potential targets for treatment of certain types of cancer[45,60]; therefore, the development of several classes of soluble claudins could accelerate drug screening and serve as a basis for the design of claudin-based biologics. Specifically in GPCR drug development, it would be transformative to create soluble analogues in specific functional states that could be used for small-molecule or antibody discovery campaigns. The precision of our design approach enabled conformational specific design for the active and inactive GPCR states, differentiated by subtle conformational changes. Consequently, our designs harboured identical G-protein-binding sites, yet they uniquely either constitutively facilitated or precluded G protein binding in solution. The computational design of specific conformational states that can mediate biological function remains an outstanding problem for which we provide a flexible and broadly applicable methodological workflow. Such an approach could constitute a basis for computational design strategies of proteins that can populate multiple conformational states in a predictable fashion, which is an important prerequisite for embedding complex functions in computationally designed proteins. From an applied perspective, the ability to create membrane-soluble analogues with native functional features could be critical in facilitating the development of new drugs and therapies that target these challenging classes of protein, which are among the most important drug targets. In summary, we present a deep learning approach for computational protein design that demonstrates the usefulness of high-quality structure representations in enabling effective exploration of new sequence spaces that can yield viable proteins and contribute to the expansion of the designable fold space, with implications for our ability to design functional proteins.

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

## Methods

### AF2$_{seq}$ design protocol

**Design target preparation.** The design target structures were sourced from the Protein Data Bank (PDB) and included the following protein folds: IGF (3SD2), BBF (6D0T)[30], TBF (5BVL)[38], claudin (4P79)[61], rhomboid protease (3ZEB)[44] and GPCR (6FFI)[62]. Owing to missing residue positions in the TBF, claudin and GPCR X-ray structures, we used AF2 to predict the protein structure using the X-ray structure as a template. Disordered regions in the claudin (residues 34–40) and GPCR (residues 875–896) targets were replaced by three-glycine and five-glycine linkers, respectively. The GPCR sequence was predicted using the experimental structure as a template but without the endolysin domain (residues 679–838) used for crystallization.

**Loss function.** For computation of error gradients, a composite loss function was used:

$$\text{loss} = W_{FAPE}L_{FAPE} + W_{dist}L_{dist} + W_{pLDDT}L_{pLDDT} + W_{pTM}L_{pTM}.$$

The loss function is represented as a combination of $L$, which denotes the value of the loss, and $W$, which denotes the weight of the loss. The frame aligned point error (FAPE) loss quantifies the L2 norm between the predicted $C_\alpha$ atoms and the target structure. The distogram (dist) loss is the cross entropy over the $C_\beta$ distogram for non-glycine residues and the $C_\alpha$ distance in the case of glycine. The model confidence (pLDDT) loss of the $C_\alpha$ positions is computed by taking $1 - \text{pLDDT}$, penalizing low confidence. Finally, the pTM score loss is a prediction confidence metric focused on global structural similarity. In this work, the designs were generated using loss terms $W_{FAPE} = 1.0$, $W_{pLDDT} = 0.2$ and $W_{pTM} = 0.2$. During initial trajectories, $W_{dist}$ was set to 0.5, whereas it was disabled during trajectory reseeding (soft starts, described below).

**Gradient descent.** As previously described in ref. 7, amino acid sequences were initialized on the basis of the secondary structure of the target fold. The secondary structure assignments were encoded in sequences, using alanines for helix, valines for β-sheet and glycines for loop residues. This introduces a bias towards the correct local structure, aiding faster convergence of the design trajectories. To diversify the generated designs, 10% of the amino acids were randomly mutated in the initial sequence of each design trajectory. Subsequently, the sequence was passed through the AF2 networks, which generated five structures. These structures were then used to calculate the loss with the previously defined loss function. The error gradient was obtained by backpropagating the errors to the one-hot-encoded input, resulting in a $5 \times 20 \times N$ error gradient, where $N$ represents the sequence length. We then took the average of the five matrices to obtain the mean error gradient ($20 \times N$), which was used for gradient descent. A position-specific scoring matrix (PSSM) of $20 \times N$ was updated using the ADAM optimizer[63] with the normalized error gradient. Following the update, the PSSM underwent a softmax function that transforms the matrix into a probability distribution of the amino acid identity for each position. The argmax function was subsequently used to determine the most probable amino acid identities per position; these were then used to construct the new input sequence for the next iteration. The cysteine residues in the PSSM were masked, so the designed sequences do not contain any cysteines.

**Model settings.** AF2 was run in single sequence mode using the network configuration of the original AF2 'model_5_ptm' for all five AF2 models with mutiple sequence alignments (MSAs) and templates disabled. For the design trajectories, we used zero recycles, meaning that each AF2 network was only executed once. For the claudin-1 and claudin-4 designs, we only used models 1 and 2 with the network configuration of the original AF2 'model_1_ptm' with templates enabled. All design runs were executed on a single Nvidia Tesla V100 (32 GB) GPU.

**Computational design protocol.** In each AF2-sequence design trajectory, 500 rounds of gradient descent optimization were performed (https://github.com/bene837/af2seq). Not all design trajectories of the claudin, rhomboid protease and GPCR converged. Hence, we sampled sequences from successful trajectories and introduced mutations, while disabling distogram loss. These sequences were then used as starting points for new design trajectories, which we named soft starts, resulting in a higher convergence rate. All generated sequences were then predicted using AF2 with three recycles, followed by relaxation in an AMBER force field[64,65]. This resulted in high-quality structures that were used as inputs to ProteinMPNN for sequence generation. The total numbers of designs and designs passing in silico filtering are summarized in Supplementary Table 1. For the design of the claudin-1 and claudin-4 functional analogues, we first predicted their structures using AF2 with MSAs and templates enabled, owing to the lack of high-resolution experimental structures. The predictions were then used as structural templates for both design and reprediction, as the wild-type extracellular region could not be predicted by AF2 in single sequence mode. All sequence and side-chain information was removed from the template to reduce folding bias. We tried several design strategies for the functional claudin design, of which two were successful: (1) redesigning only the transmembrane surface, approximately 40% of the sequence; and (2) redesigning the entire transmembrane region, including the core, approximately 60% of the sequence. The residue positions that were fixed can be found in Supplementary Table 2.

For conformation-specific design of GPCRs, we used the template of the adenosine A2A GPCR in the active conformation bound to mini-G$_s$ (PDB 5G53) and the inactive conformation (PDB 3VGA) to design each state individually. We fixed residues interacting with the G protein and the evolutionarily conserved DRY motif during the design of each state, resulting in designs with identical length and identical functional sites. For the design of the active conformation, we found that it was not possible to generate high-confidence designs without the presence of G protein; hence, gradient descent and prediction were performed in the presence of the mini-G$_s$ binder.

### Training of MPNN$_{sol}$

The MPNN$_{sol}$ model was trained on protein assemblies in the PDB (as of 2 August 2021) determined by X-ray crystallography or cryo-EM to a resolution of better than 3.5 Å and with fewer than 10,000 residues. We followed training as described in ref. 13, modified only by excluding annotated transmembrane PDB codes. The list of excluded PDB codes and MPNN$_{sol}$ model weights are available at https://github.com/dauparas/ProteinMPNN/tree/main/soluble_model_weights.

### ProteinMPNN sequence redesign

The backbones generated by AF2$_{seq}$ were used as inputs to Protein-MPNN. For the vanilla ProteinMPNN, we used the provided model weights trained on a dataset with 0.1 Å Gaussian noise[13]. For the biased ProteinMPNN (referred to in the main text as MPNN$_{bias}$), we used a modified version of the script 'submit_example_8.sh' as provided on the ProteinMPNN github mentioned above. We found the best results by giving a positive sampling bias to the polar amino acids and a negative sampling bias to alanine. For MPNN$_{sol}$, we generated sequences with two different models that had different levels of noise during training (0.1 Å and 0.2 Å). For all ProteinMPNN models, we generated two sequences per AF2$_{seq}$-designed backbone. No Gaussian noise was added to the input backbone, and cysteine residues were masked during the decoding process.

## Structural similarity calculations

The $C_\alpha$ atoms of the structures were aligned using the Superimposer from the Biopython package[66]. The r.m.s.d.$_{C\alpha}$ was calculated as the mean Euclidean distance between predicted and target $C_\alpha$ atom coordinates. The r.m.s.d.$_{fa}$ was calculated by first aligning all atoms with the Superimposer, after which the mean Euclidean distance between atoms was computed. The template modelling scores were determined using TM-align[67].

## Sequence diversity analysis

Sequence recovery was quantified as the number of positions at which the corresponding residue matches the residue in the target fold divided by the total number of residues in the sequence multiplied by 100%. The core residues were defined as residues with less than 20 Å$^2$ solvent-accessible surface area (SASA), and surface residues were defined as residues with less than 20 Å$^2$ SASA. The $e$-values were obtained through a protein BLAST search of the NCBI RefSeq database of 1 October 2022 with a maximum hit value of 1,000.

## Surface hydrophobicity calculations

The fraction of surface hydrophobics was calculated using Rosetta[3]. First, all surface residues were identified using the layer selector; these were defined as residues with SASA > 40 Å$^2$. Of these surface residues, we counted the number of apolar amino acids (defined as 'GPAVILM-FYW') and divided it by the total number of surface residues.

## Design filtering and selection

All generated sequences were predicted with AF2 using three recycles and a relaxation step in an AMBER force field. Next, the sequences were filtered using the following criteria: (1) TM score > 0.80 for all designs except the rhomboid protease (the rhomboid protease yielded slightly lower TM scores in the design trajectory; hence, we chose a cut-off value 0.75 instead); (2) pLDDT > 80 for all designs except the rhomboid protease (pLDDT > 75); and (3) an $e$-value threshold > 0.1 for sequence novelty. Success rates are listed in Supplementary Table 1.

## Structural fold similarity search

The fold similarity search was performed using FoldSeek[68] on the SCOP database[17] (downloaded March 2023). For each of the design target folds, an exhaustive search on the basis of TM score alignment was performed. The SCOP database contains globular and membrane domain annotations, which were used for the hit classification.

## Fold complexity calculations

Relative contact order was calculated at the secondary structure level by computing the residue distance in the sequence between secondary structures for all pairs within 8 Å of each other and then averaging these distances for all contacts that were more than four residues apart. To ensure consistency in secondary structure annotations across all structures, we used DSSP for the determination of secondary structural elements[17]. The de novo protein dataset comprised 70 helical proteins, six β-sheet proteins and 42 proteins containing both α-helices and β-sheets[34,69]. The natural protein dataset consisted of 1,000 proteins randomly selected from the entire collection of proteins in the CATH dataset (v.4.3)[70].

## Transplantation of natural epitopes on to soluble scaffolds

Compatible epitopes were identified by means of a Foldseek search of the PDB, using soluble scaffolds as queries. Hits with TM scores above 0.7 and high structural similarity around the desired epitope were superimposed using structure visualization software, such as PyMOL or ChimeraX. Varying lengths of the epitope were selected for transplantation, encompassing either only interaction sites, entire loops or overlapping parts of the supporting secondary structures. The sequence of the overlaid epitope was then pasted into the overlapping region of interest in the soluble scaffold. The resulting chimeric sequences were predicted using AF2 in single sequence mode. Structures with high pLDDT (greater than 90) and high TM scores relative to the starting scaffold were manually inspected to verify the placement of the epitope. Finally, a subset of constructs in different soluble scaffolds were selected for experimental testing.

## SPR binding assay

SPR measurements were carried out on a Biacore 8K system (Cytiva) in HBS-EP+ buffer (10 mM HEPES pH 7.4, 150 mM NaCl, 3 mM EDTA, 0.005% (v/v) Surfactant P20 Cytiva). The antibody (5 µg ml$^{-1}$) was immobilized on a CM5 sensor chip (Cytiva) by amide coupling in 10 mM NaOAc pH 4.5 (250 s, 10 µl min$^{-1}$; 700–1500 response units immobilized). Purified mini-G$_s$ was immobilized with a contact time of 200 s (300 response units immobilized). Binding assays were carried out at a flow rate of 30 µl min$^{-1}$. Designed chimeras were injected as serial dilutions ranging from 18 µM to 0.1 nM, and 0 nM for 120 s, followed by dissociation for 400 s. Immobilized antibody was regenerated between cycles in 10 mM glycine-HCl pH 2.5 (30 s, 30 µl min$^{-1}$). GPCRs designed in the active or inactive state were injected at 0, 5, 15 and 25 µM for 90 s, followed by dissociation for 120 s. Immobilized mini-G$_s$ ligand was not regenerated between cycles. Binding curves were fitted with a 1:1 Langmuir binding model in the Biacore 8K analysis software. Steady-state response units were plotted against analyte concentration, and a sigmoid function was fitted to the experimental data in Python 3.9 to derive the $K_d$.

## Bio-layer interferometry

For BLI studies of claudins, synthetic claudin-His and tagless CpE in 20 mM Tris pH 7.4, 100 mM NaCl, and 5% glycerol were used. BLI was performed at 25 °C in 96-well black flat-bottomed plates (Greiner) using an acquisition rate of 5 Hz averaged by 20 using an Octet R8 System (FortéBio/Sartorius), with assays designed and set up using Blitz Pro 1.3 software. Binding experiments consisted of the following steps: sensor equilibration (30 s), loading (300 s), baseline (180 s), and association and dissociation (120–300 s each). Experiments were conducted by immobilizing 1.5–3.0 µM of synthetic claudin-His on NiNTA (Dip and Read) sensors and quantifying their binding to 0.05–5.00 µM CpE. Association and dissociation times for the two claudin-1 designs were performed for 120 s, as they exhibited rapid on and off rates, whereas for the claudin-4 design, these times were extended to 300 s to capture the slower off rates. Data were fitted to a 1:1 binding model using Octet Analysis Studio (Sartorius), which generated the $K_d$ from the association and dissociation rate constants. At the protein concentrations used, no significant non-specific binding of CpE to NiNTA sensors was detected.

## Protein crystallization and structure determination

The TBF_24 design was crystallized using sitting drop vapour diffusion at 4 °C in 0.1 M Na$_3$ citrate pH 4.0, 1 M LiCl, and 20% PEG 6000 buffer. The CLF_4 design was crystallized using sitting drop vapour diffusion at 4 °C in 0.1 M Na$_3$ citrate pH 5.0, 0.1 M Na/K phosphate pH 5.5, 0.1 M RbCl, and 25% v/v PEG smear medium (BCS Screen, Molecular Dimensions). The RPF_9 design was crystallized using sitting drop vapour diffusion at 4 °C in 0.1 M HEPES pH 7.8, 0.15 M Na$_3$ citrate dihydrate, and 25% v/v PEG smear low (BCS Screen, Molecular Dimensions). The GLF_18 design was crystallized using sitting drop vapour diffusion at 4 °C in Na phosphate-citrate pH 4.2, 0.2 M LiSO$_4$, and 20% PEG 1000 buffer. The GLF_32 design was crystallized using sitting drop vapour diffusion at 4 °C in 0.1 M Na acetate pH 5.5, 0.2 M KBr, and 25% PEG MME 2000 buffer. Crystals were cryoprotected in 20% glycerol and flash-cooled in liquid nitrogen. Diffraction data were collected at the beamline PXI (X06SA) of the Swiss Light Source (Paul Scherrer Institute, Villigen, Switzerland) and the MASSIF-1 beamline of the European Synchrotron Radiation Facility (Grenoble, France) at a temperature of 100 K. Data were processed using the autoPROC package[71]. Phases

were obtained by molecular replacement using Phaser[72]. Atomic model refinement was completed using COOT[73] and Phenix.refine[72]. The quality of refined models was assessed using MolProbity[74]. Structural figures were generated using PyMOL (Schrödinger, LLC; https://www.pymol.org/) and ChimeraX[75]. Data collection and refinement statistics are listed in Extended Data Table 1.

### Cryo-EM structure determination of CLN4-20 in complex with cCpE

Expression and purification of cCpE, COP-2 Fab and the anti-Fab nanobody were performed as described previously[76]. Concentrated CLN4_20 was complexed with cCpE followed by COP-2 in a 1:1.2:1 molar excess. Next, the anti-Fab nanobody was added at a 1.3 molar excess of COP-2, followed by incubation on ice for 30 min, concentrated and subjected to SEC using a Superdex 200 increase 10/300 GL column (GE Healthcare) in 20 mM HEPES pH 8.0, 150 mM NaCl. The purified complex was concentrated to 5 mg ml$^{-1}$.

UltraAuFoil 1.2/1.3 grids (Quantifoil) were glow discharged for 30 s at 15 mA and vitrified using a Leica GP2 instrument (Leica microsystems). Then, 3.5 μl of the complex was applied to grids and blotted for 3 s at 4 °C under 100% humidity, before being plunge frozen into liquid ethane. Grid screening and data collection were performed on a 200 kV Glacios 2 Cryo-TEM (ThermoFisher Scientific) with a Falcon 4i direct electron detector at Hauptman-Woodward Medical Research Institute. A total of 1,159 videos were collected at a physical pixel size of 0.884 Å, with an electron dose of 49.4 e/Å$^2$ fractioned over 93 frames.

Videos were processed, patch motion corrected and patch CTF estimated in cryoSPARC. Blob picking generated a suitable template for an initial three-dimensional volume; this was used to produce two-dimensional projections for template picking, followed by two-dimensional classification, ab initio reconstruction and three-dimensional refinement, resulting in a cryo-EM density resolved to a resolution of 4.1 Å. Structural coordinates for the complex of CLN4_20, cCpE and COP-2 Fab from PDB ID 7TDM[76] were rigid body docked. The nanobody from PDB 8U4V was docked on to the L chain of COP-2. Each protein chain was then real-space refined in Coot. Final model refinement was conducted with Namdinator[77], followed by real-space refinement using Phenix phenix.real_space_refine[72]. Extended Data Table 2 shows data collection and refinement statistics for the CLN-4_20/cCpE/COP-2/Nb structure.

### Reporting summary

Further information on research design is available in the Nature Portfolio Reporting Summary linked to this article.

### Data availability

All data are available in the paper and its Supplementary Information. Atomic coordinates and structure factors of the reported X-ray structures have been deposited in the PDB under accession numbers 8OYS (TBF_24), 8OYV (CLF_4), 8OYW (RPF_9), 8OYX (GLF_18) and 8OYY (GLF_32). Atomic coordinates and cryo-EM density of the CLN-4_20/cCpE/COP-2/Nb complex have been deposited in the PDB under accession number 9BEI and in the Electron Microscopy Data Bank with entry number 44479. Source data are provided with this paper.

### Code availability

Af2seq code is available at GitHub (https://github.com/bene837/af2seq). AF2 model weights used for design and predictions can be downloaded from https://storage.googleapis.com/alphafold/alphafold_params_2021-07-14.tar. ProteinMPNN, with soluble trained weights, is available at GitHub (https://github.com/dauparas/ProteinMPNN).

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

**Acknowledgements** We thank SCITAS at Ecole Polytechnique Fédérale de Lausanne (EPFL) for support in running design trajectories; B. Blattmann (Protein Crystallization Center, UZH, Zurich, Switzerland) for performing initial crystallization screens; F. Pojer, K. Lau and A. Larabi (Protein Production and Structure Characterization Core Facility, EPFL, Switzerland) for help with crystal optimization and data processing; T. Tomizaki (Swiss Light Source, X06SA beamline, Paul Scherrer Institute, Villigen, Switzerland) and D. Nurizzo (European Synchrotron Radiation Facility, MASSIF-1 beamline, Grenoble, France) for assistance with crystallographic data collection; and C. Tate (MRC Laboratory of Molecular Biology, Cambridge) for donating the expression plasmid for mini-G$_s$-414. A.J.V. acknowledges that cryo-EM experiments were conducted at the Hauptman Woodward Institute cryo-EM Center and thanks A. Kossiakoff at the University of Chicago for developing the synthetic antibody fragment library that yielded the Fab used in the cryo-EM studies presented here. M.P. was supported by the Peter und Traudl Engelhorn Stiftung. B.E.C. was supported by the Swiss National Science Foundation, the National Centre of Competence in Research (NCCR) in Chemical Biology, the NCCR in Molecular Systems Engineering and ERC starting grant no. 716058. J.D. was supported by a gift from Microsoft, and D.B. was supported by the Howard Hughes Medical Institute. S.O. and S.K. were supported by NIH DP5OD026389, NSF MCB2032259 and Amgen. A.J.V. was supported by the National Institute of General Medical Sciences of the National Institutes of Health under award number R35GM138368.

**Author contributions** C.A.G., M.P. and B.E.C. conceived the study and designed experiments. C.A.G., J.D. and S.O. developed the code base. C.A.G., M.P. and N.G. generated protein designs. M.P., N.G., L.J.D., P.E.M.B., S.G., S.R., S.K. and J.C. purified proteins and performed experimental characterization. M.P. solved crystal structures. S.K., J.C. and A.J.V. solved cryo-EM structures. C.S. performed molecular dynamics perturbations. C.A.G. and S.K. performed fold complexity analysis. J.D. and D.B. trained MPNN$_{sol}$. S.O., A.J.V., D.B. and B.E.C. supervised the work and acquired core funding. C.A.G., M.P., N.G. and B.E.C. wrote the initial manuscript. All authors read and contributed to the manuscript. C.A.G., M.P. and N.G. agree to rearrange the order of their respective names according to their individual interests.

**Competing interests** The authors declare no competing interests.

**Additional information**
**Correspondence and requests for materials** should be addressed to Bruno E. Correia.

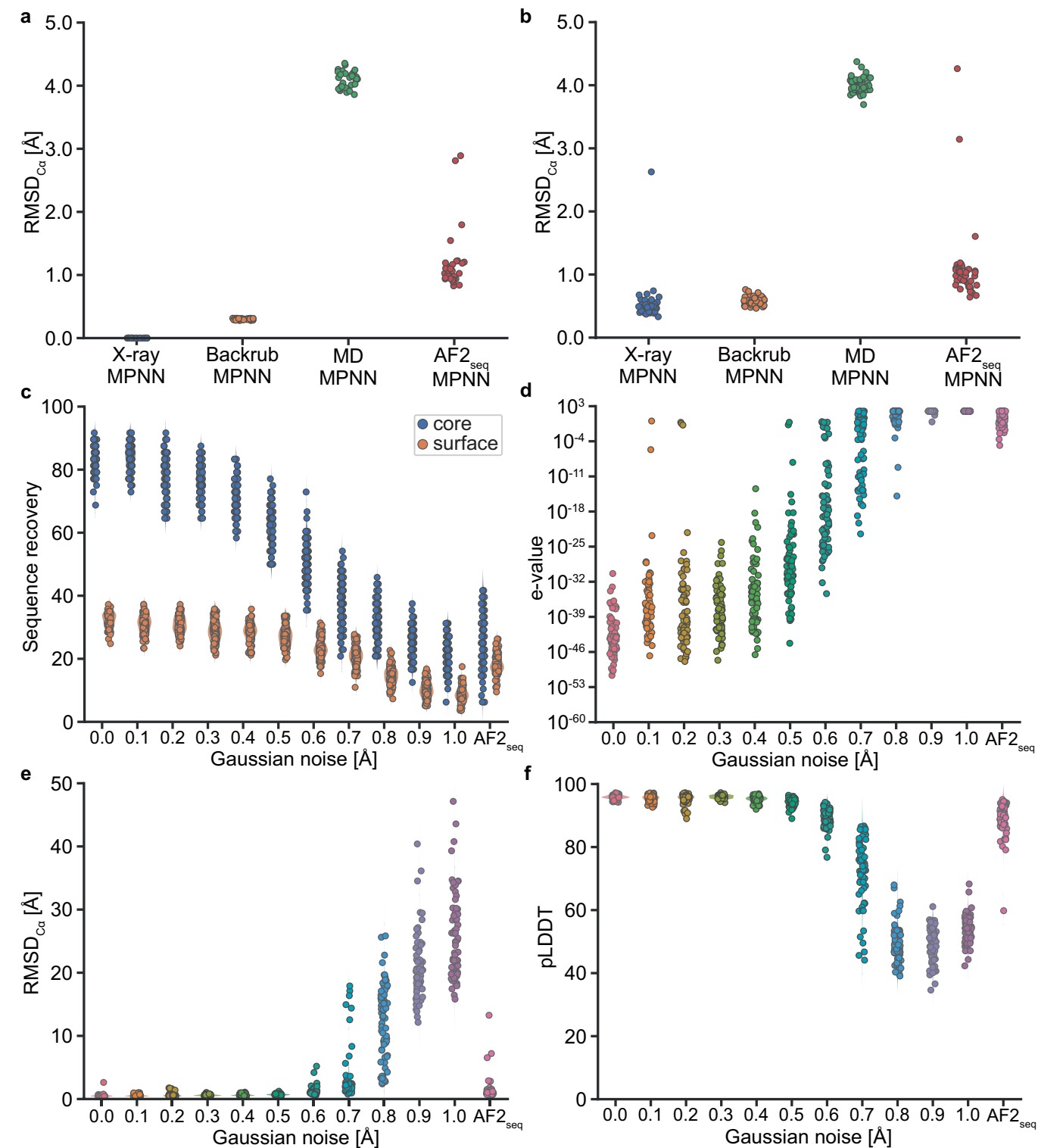

**Extended Data Fig. 1 | RMSD$_{C\alpha}$ of designed TIM-barrel structures vs the target fold crystal structure (PDB ID: 5BVL). a**, Backbone RMSD$_{C\alpha}$ deviations of input structures used for ProteinMPNN sequence redesign. **b**, Backbone RMSD$_{C\alpha}$ deviations of the highest ranked AF2 predicted structure derived from the ProteinMPNN-designed sequences from panel a. **c**, Sequence recovery percentage by ProteinMPNN in the core and on the surface with different values of Gaussian noise applied to the backbone atoms. **d**, The e-values of the generated ProteinMPNN sequences with varying degrees of noise compared to AF2$_{seq}$ generated sequences. **e**, Backbone RMSD$_{C\alpha}$ deviations of predicted structures of ProteinMPNN and AF2$_{seq}$ generated sequences. **f**, AF2 confidence (pLDDT) scores of predicted structures.

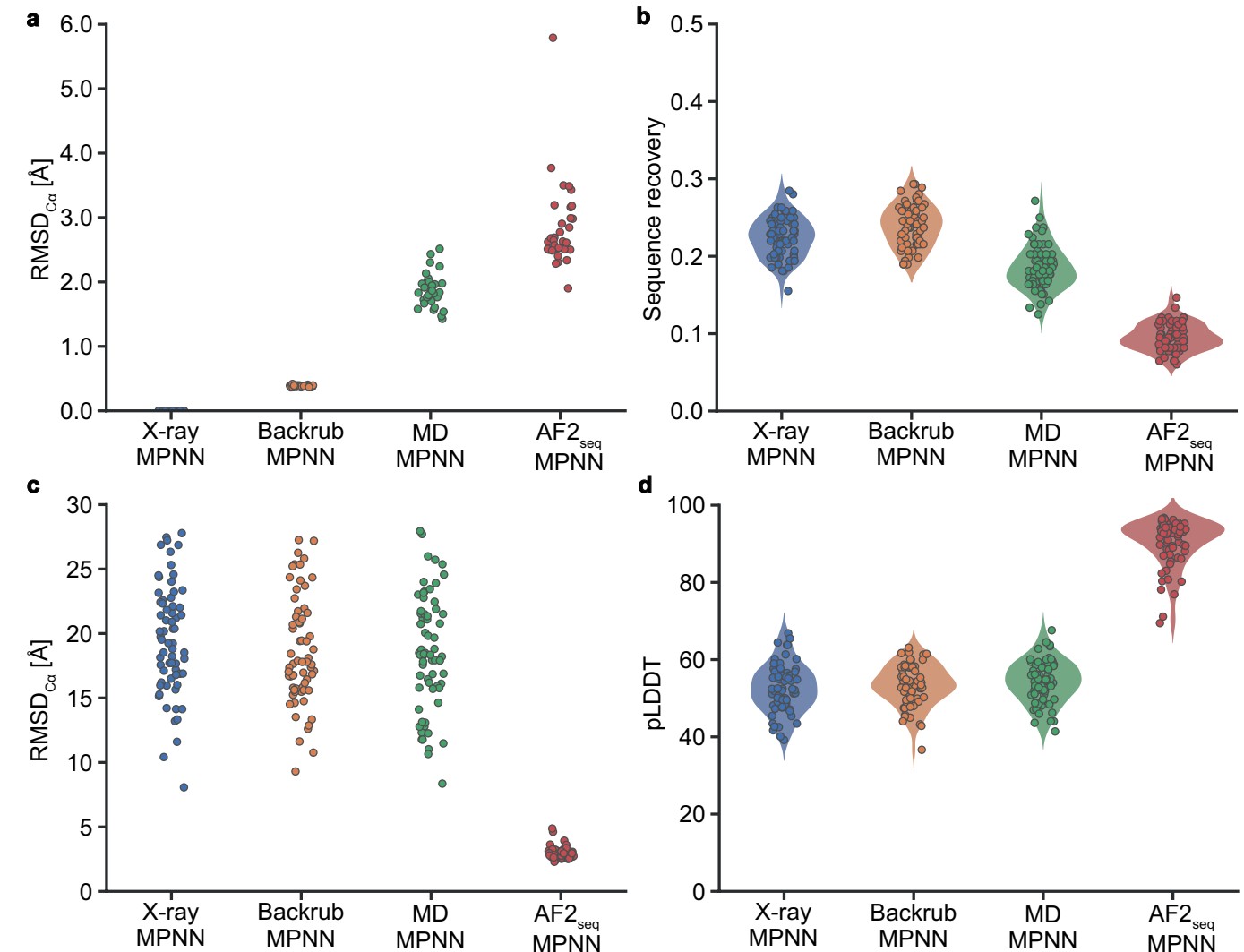

**Extended Data Fig. 2 | Computational analysis of GPCR backbone (PDB ID: 6FFI) perturbation methods for diversifying sequence design. a**, Backbone $RMSD_{C\alpha}$s relative to the reference crystal structure after being perturbed using Rosetta backrub, MD simulations, or using our $AF2_{seq}$ pipeline. **b**, Sequence recovery rates of ProteinMPNN sequence generation on the perturbed backbones. **c**, Backbone $RMSD_{C\alpha}$ of the top ranking AF2 model of ProteinMPNN sequences relative to the reference crystal structure. **d**, AF2 confidence scores of top ranking ProteinMPNN-derived predictions.

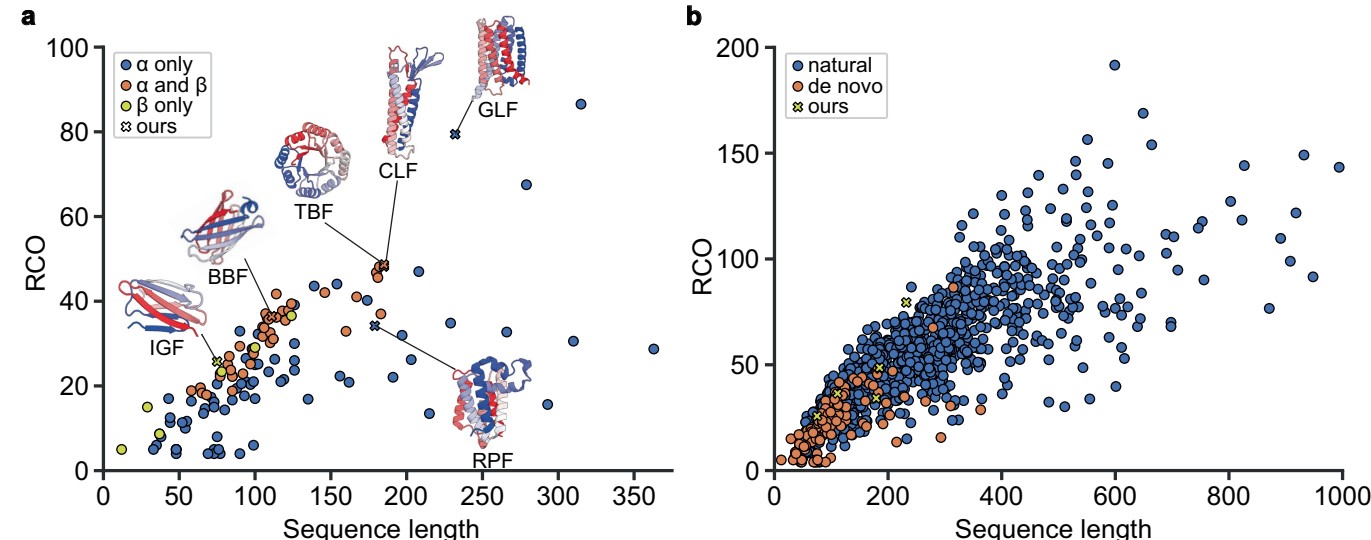

**Extended Data Fig. 3 | Relative Contact Order (RCO) plotted against sequence length of de novo designed and natural proteins.** Both RCO and sequence length describe the complexity of a protein (see Methods). This metric quantifies the number of contacts in a protein structure dependent on the sequence separation in order to capture the nonlocality in sequence of those contacts. **a**, Curated set of structures from computationally designed proteins reported by Verkuil et al. [34] and Woolfson[69] (shown in circles) were compared to the design targets in this paper (shown in crosses). This assessment shows that many of the designed topologies show high contact orders relative to other computationally designed proteins previously reported. Symbols are colored according to **b**, Comparison with natural folds shows that in general native proteins have higher contact orders then computationally designed proteins.

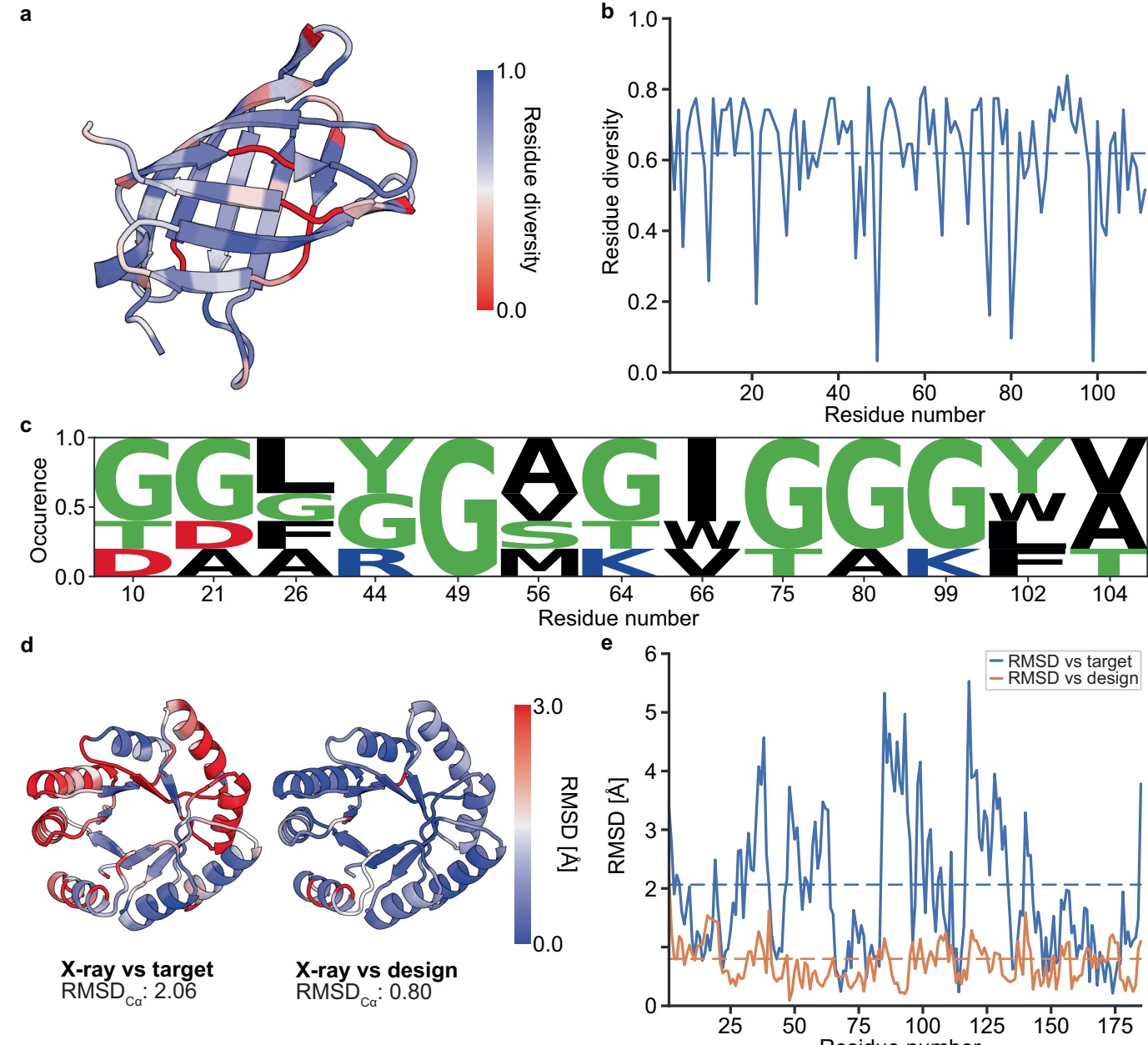

**Extended Data Fig. 4 | Sequence and structural conservation analysis of β-barrel (BBF) and TIM-barrel folds (TBF). a**, Cartoon depiction of example BBF design colored by sequence diversity of all designs on an individual residue level. **b**, Sequence diversity plotted on a per-residue level of all BBF designs. The dotted line represents the mean sequence diversity of the structure. **c**, Sequence logo of residue occurrence of experimentally validated and folded

BBF designs highlighting residue variability at sites critical for maintaining β-barrel topology. **d**, Cartoon depiction of the crystal structure of TBF_24 colored by backbone $RMSD_{C\alpha}$ per residue when compared against the target fold (left) and the design model (right), with $RMSD_{C\alpha}$ per residue values plotted in panel **e**.

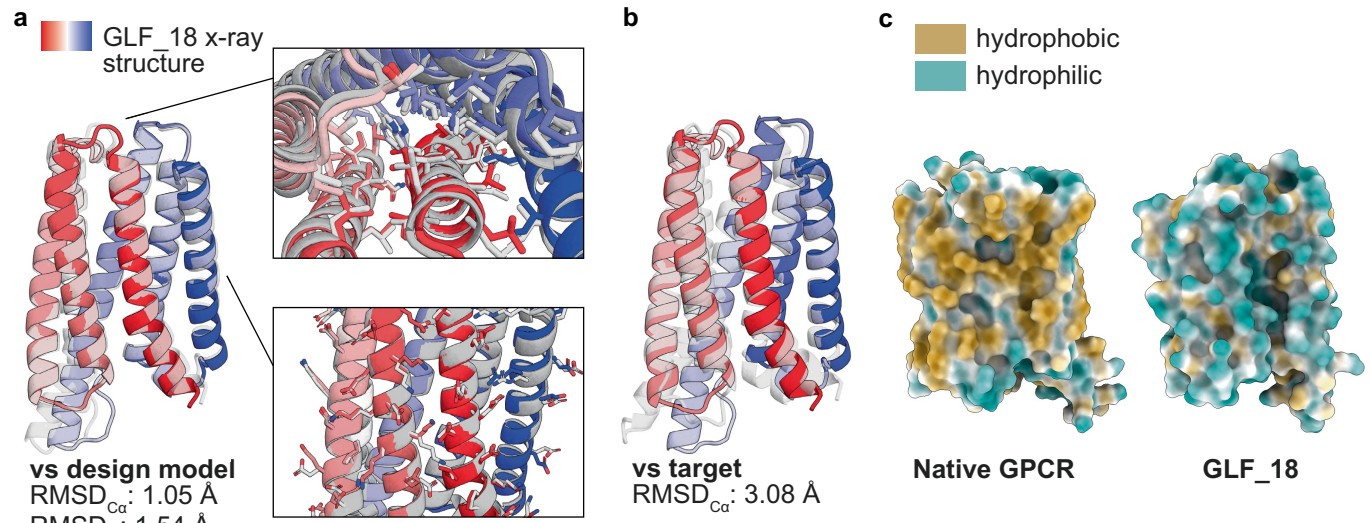

**a** GLF_18 x-ray structure

**vs design model**
RMSD$_{C\alpha}$: 1.05 Å
RMSD$_{fa}$: 1.54 Å

**b** **vs target**
RMSD$_{C\alpha}$: 3.08 Å

**c** hydrophobic
hydrophilic

**Native GPCR**          **GLF_18**

**Extended Data Fig. 5 | GLF_18 X-ray structure. a**, X-ray structure of GLF_18 (colored) superimposed on the design model (gray). **b**, X-ray structure of GLF_18 (colored) superimposed on the design model (gray). **c**, Molecular lipophilicity potential of the surface of the GPCR design target and the soluble GLF_18 design.

RMSD$_{C\alpha}$ - root mean square deviation computed over the Cα atoms of the backbone. RMSD$_{fa}$ - root mean square deviation computed over all the atoms in the structure.

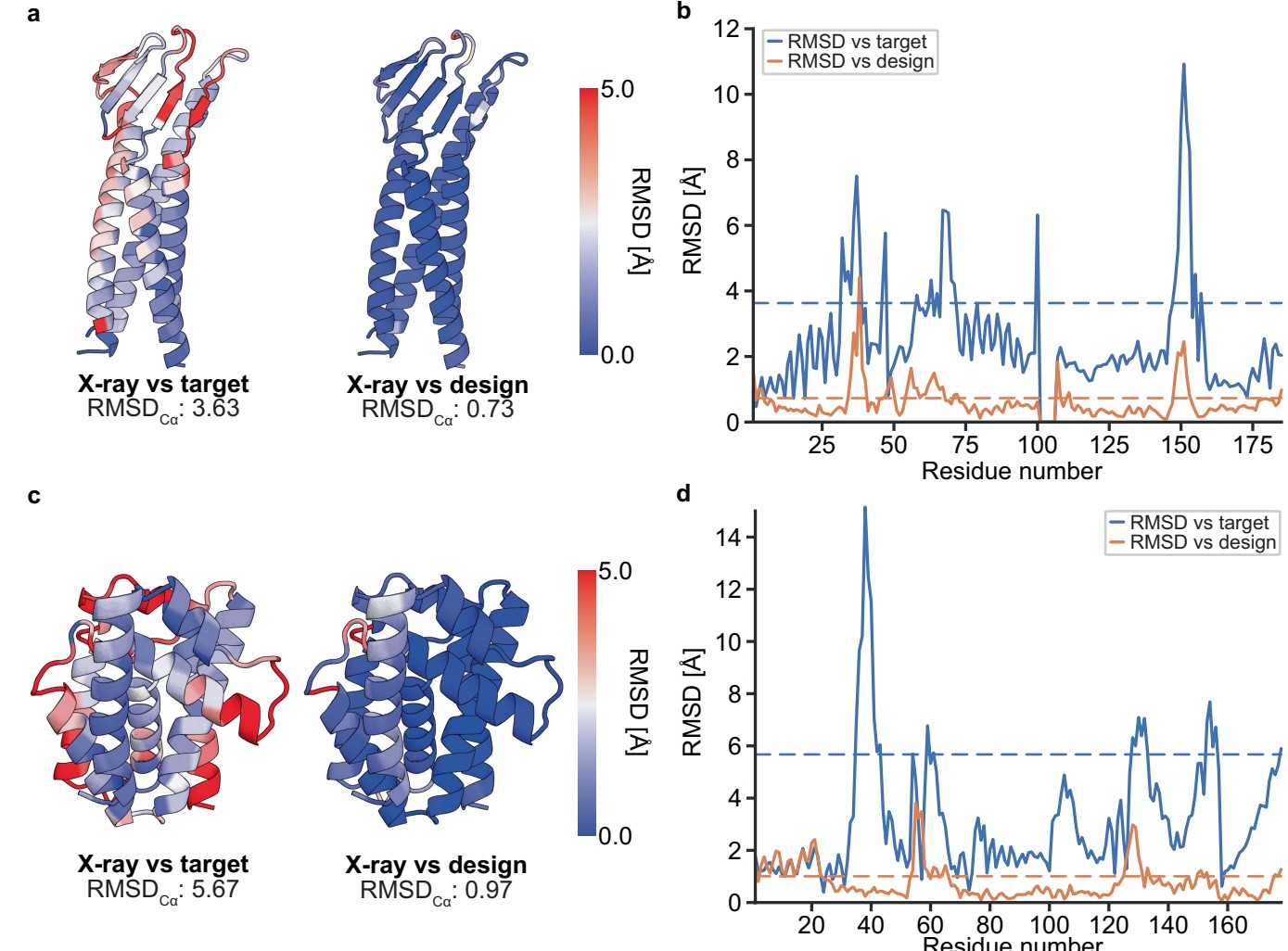

**Extended Data Fig. 6 | Backbone RMSD$_{C\alpha}$s of CLF_4 and RPF_9 crystal structures relative to the target fold and design model. a**, Cartoon representation of the crystal structure of CLF_4, with the left and right models colored by backbone RMSD$_{C\alpha}$ per residue when compared against the target fold and the design model, respectively. **b**, RMSD$_{C\alpha}$ per residue values for both comparisons. In the comparison with the target fold the largest differences are found in the β-sheet region. **c**, Cartoon representation of the crystal structure of RPF_9, with the left and right structures colored by backbone RMSD$_{C\alpha}$ per residue when compared against the target fold and the design model, respectively. **d**, RMSD$_{C\alpha}$ per residue values for both comparisons. The loop region between residues 35 and 42 was not structurally similar between the RPF_9 design and the target fold, however, the X-ray structure closely matches the designed model.

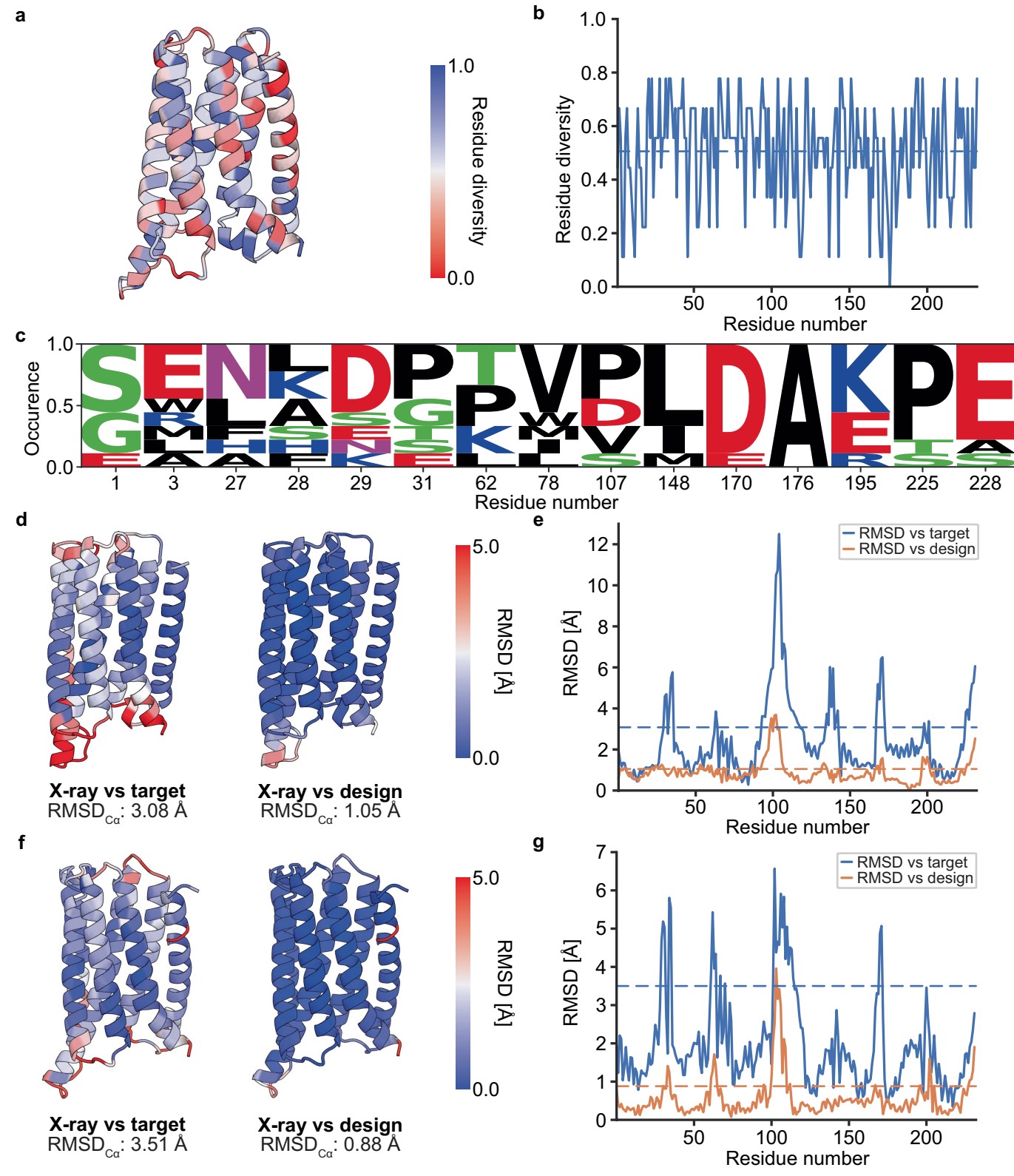

**Extended Data Fig. 7 | Sequence and structural conservation analysis of GPCR-like folds (GLF). a**, Cartoon representation of a GLF colored by residue diversity of the in vitro validated folded designs. **b**, The sequence diversity of all GLF designs is visualized on a per-residue level in the plot, with a dotted line indicating the average sequence diversity across the structure. **c**, Sequence logo of residues in experimentally validated and folded GLF designs. The natural GLF contained a highly conserved DRY motif in the first intracellular loop (residues 26 to 28) and a PXXY motif (residues 225 to 227) in the seventh

helix which are not present in our designs. All other positions depicted contained proline residues in the design target, which have a higher prevalence for some positions but are not required. **d**, Cartoon depiction of the crystal structure of GLF_18 colored by backbone $RMSD_{C\alpha}$ per residue when compared against the target fold (left) and the design model (right), with individual values plotted in panel **e**. **f**, Depiction of the GLF_32 crystal structure colored by backbone $RMSD_{C\alpha}$, with individual values plotted per residue in panel **g**.

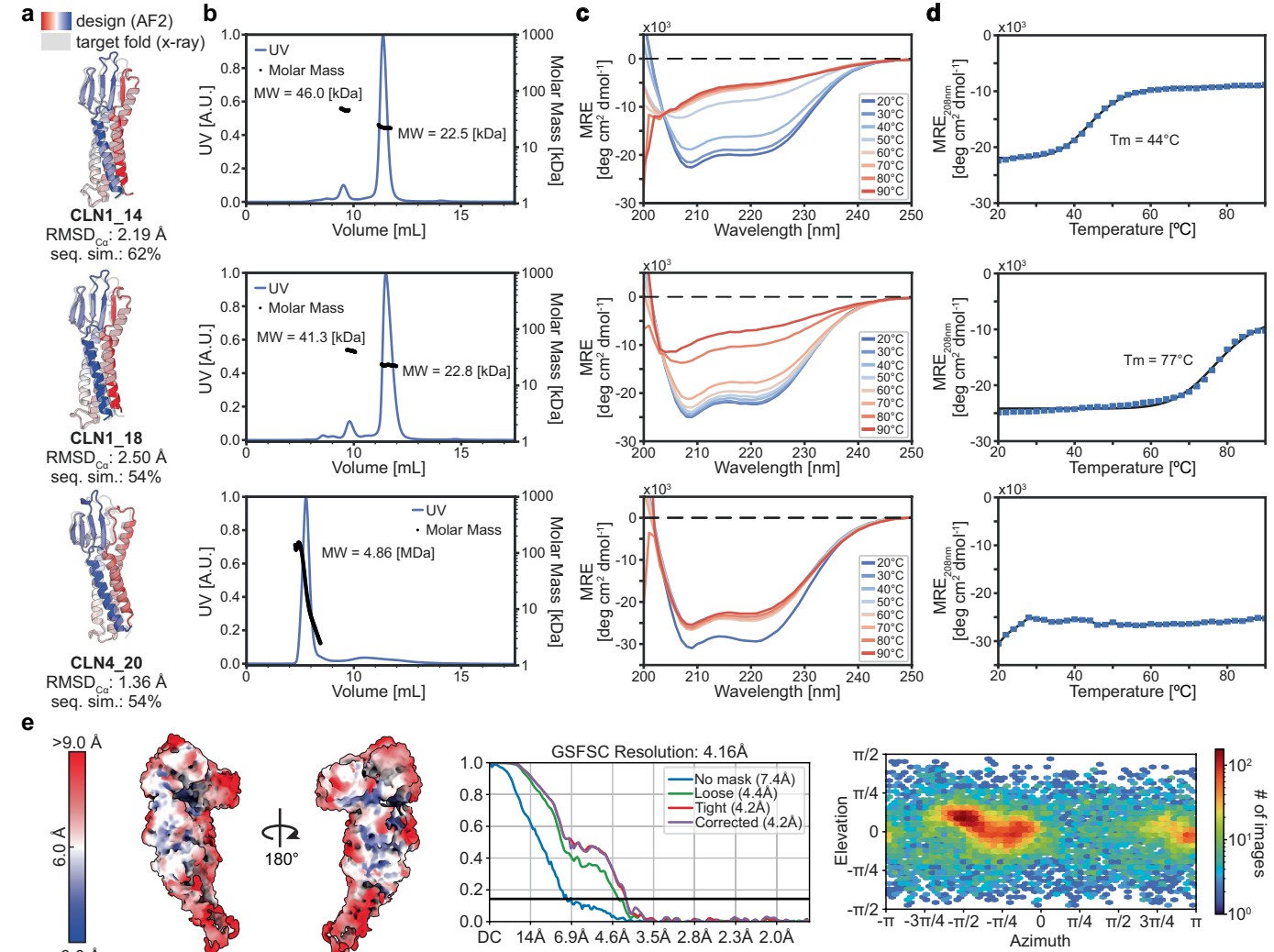

**Extended Data Fig. 8 | Characterization of the solubilized human Claudin-1 (CLN1) and Claudin-4 (CLN4) presenting functional motifs. a**, Cartoon depiction of design (colored) overlaid on the target fold (gray). **b**, SEC-MALS analysis of corresponding design in panel a. The expected Mw for the monomeric design ranges from 27.2 to 28.1 kDa. **c**, CD spectroscopy measurements at different temperatures. **d**, Thermostability curve based on CD measurements. **e**, Left, CLN4_20-cCpE-COP2-nanobody complex unsharpened cryoEM map used for model docking colored by local resolution. Middle, Gold-standard FSC curve with resolution cutoff indicated at 0.143. Right, Particle distribution heatmap of the final reconstruction.

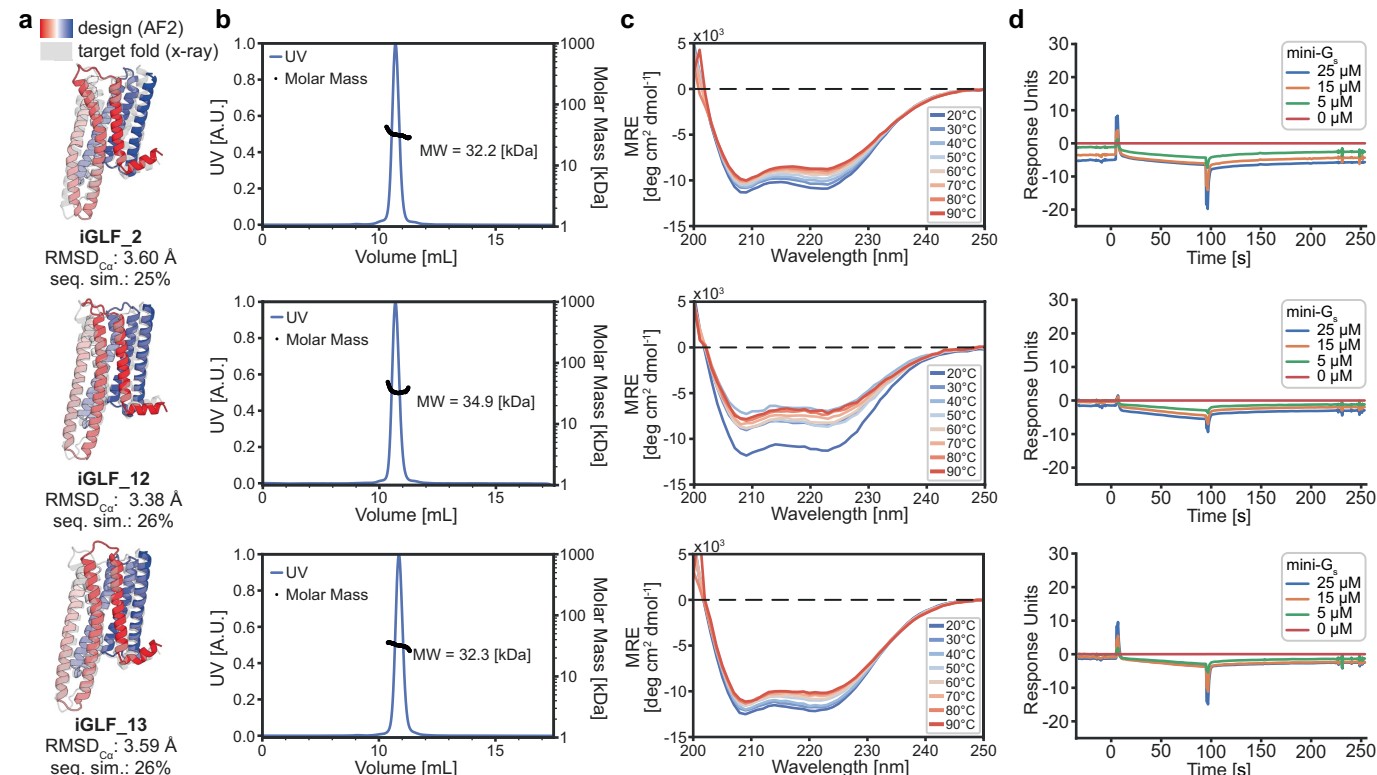

**Extended Data Fig. 9 | Biophysical characterization of the inactive state GLFs containing G-protein binding sites. a**, Cartoon depiction of design (colored) overlaid on the target fold (gray). **b**, SEC-MALS analysis of corresponding design in panel a. The expected Mw for the monomeric design ranges from 33.3 to 33.7 kDa. **c**, SPR sensorgram of different MiniG$_s$ concentrations in the presence of the designs from panel **a**.

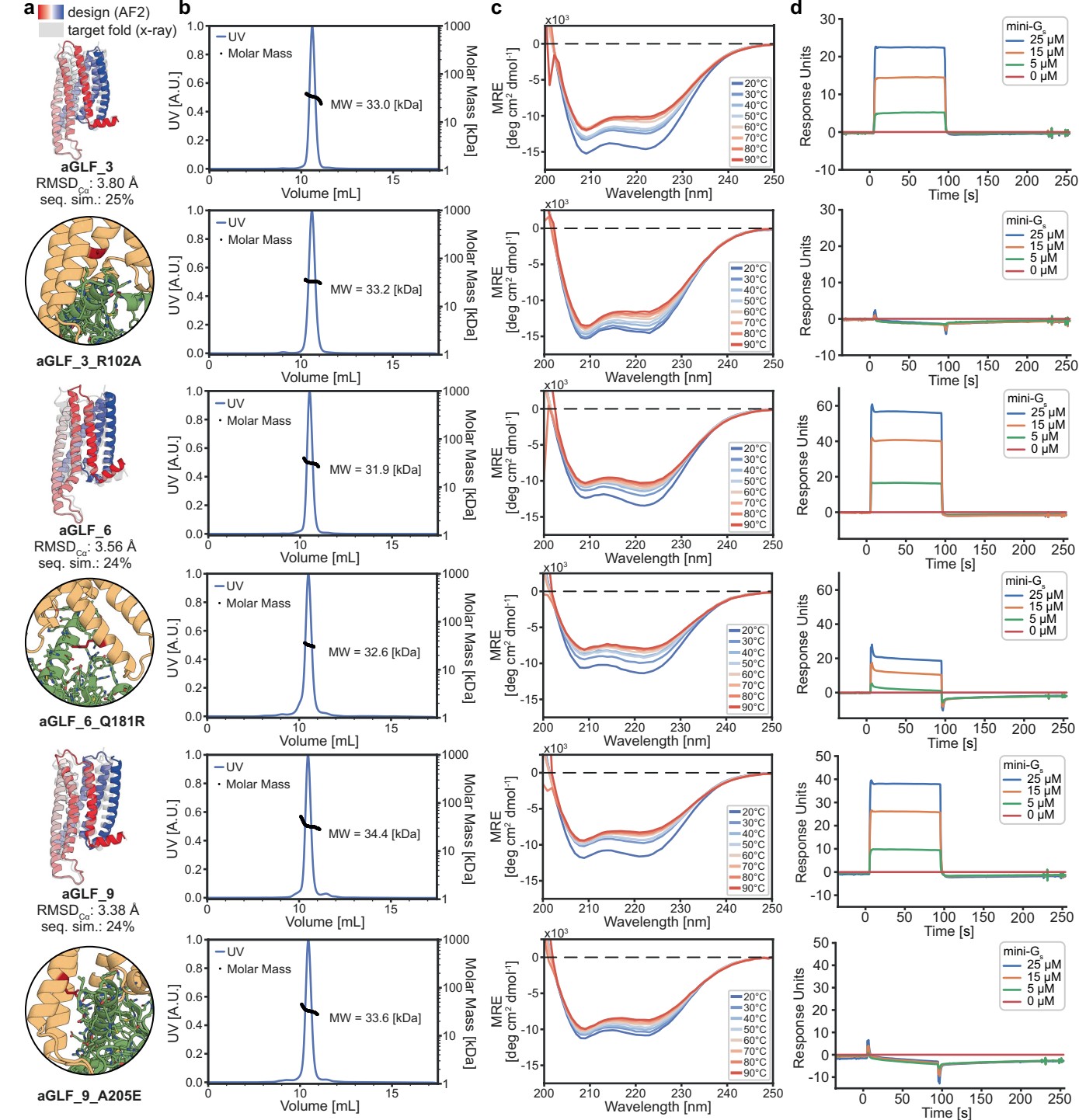

**Extended Data Fig. 10 | Biophysical characterization of the active state GLFs containing G-protein binding sites. a**, Cartoon depiction of design (colored) overlaid on the target fold (gray). The point mutants of the aGLF (orange) show a zoom with the point mutation (red) in the presence of mini-g$_s$ (green). **b**, SEC-MALS analysis of corresponding design in panel a. The expected Mw for the monomeric design ranges from 33.1 to 34.1 kDa. **c**, SPR sensorgram of different MiniG$_s$ concentrations in the presence of the designs from panel **a**.

**Extended Data Table 1 | Crystallographic data collection and refinement statistics (molecular replacement)**

| | TBF_24 (PDB: 8OYS) | CLF_4 (PDB: 8OYV) | RPF_9 (PDB: 8OYW) | GLF_18 (PDB: 8OYX) | GLF_32 (PDB: 8OYY) |
|---|---|---|---|---|---|
| **Data collection** | | | | | |
| Space group | P $2_1 2_1 2_1$ | P 1 | P $2_1 2_1 2_1$ | P 1 | P $2_1 2_1 2_1$ |
| Cell dimensions | | | | | |
| $a, b, c$ (Å) | 36.63, 44.41, 125.46 | 35.56, 50.77, 54.93 | 39.72, 66.68, 73.39 | 37.73, 52.08, 58.39 | 39.28, 56.78, 202.97 |
| $\alpha, \beta, \gamma$ (°) | 90.00, 90.00, 90.00 | 107.61, 90.50, 90.06 | 90.00, 90.00, 90.00 | 94.08, 104.36, 110.80 | 90.00, 90.00, 90.00 |
| Resolution (Å) | 62.73 - 1.34 (1.39 - 1.34) | 48.39 - 2.80 (2.90 - 2.80) | 34.93 - 1.50 (1.55 - 1.50) | 39.97 - 2.11 (2.19 - 2.11) | 49.55 - 1.85 (1.92 - 1.85) |
| $R_{merge}$ | 0.058 (1.422) | 0.1069 (1.588) | 0.0805 (1.196) | 0.050 (0.335) | 0.096 (1.526) |
| $I / \sigma I$ | 15.6 (1.4) | 6.34 (1.42) | 11.8 (1.5) | 9.9 (3.4) | 15.4 (1.7) |
| Completeness (%) | 99.9 (100.0) | 88.0 (95.1) | 99.6 (99.7) | 88.5 (92.8) | 97.7 (99.8) |
| Redundancy | 9.2 (9.5) | 2.5 (2.7) | 8.1 (7.5) | 1.7 (1.7) | 11.6 (11.6) |
| | | | | | |
| **Refinement** | | | | | |
| Resolution (Å) | 62.73 - 1.34 | 48.39 - 2.80 | 34.93 - 1.5 | 39.97 - 2.11 | 49.55 - 1.85 |
| No. reflections | 46880 (4599) | 7983 (858) | 31846 (3158) | 20185 (2144) | 38882 (3924) |
| $R_{work} / R_{free}$ | 0.1673/0.1885 | 0.2725/0.3168 | 0.1967/0.2159 | 0.2021/0.2500 | 0.1936/0.2379 |
| No. atoms | | | | | |
| Protein | 1390 | 2950 | 1502 | 3686 | 3749 |
| Ligand/ion | 2 | 0 | 1 | 10 | 6 |
| Water | 232 | 1 | 163 | 44 | 203 |
| $B$-factors | | | | | |
| Protein | 28.5 | 97.7 | 38.6 | 53.5 | 41.1 |
| Ligand/ion | 46.8 | - | 55.1 | 74.7 | 58.4 |
| Water | 39.1 | 74.0 | 43.2 | 49.2 | 42.4 |
| R.m.s. deviations | | | | | |
| Bond lengths (Å) | 0.006 | 0.002 | 0.008 | 0.006 | 0.006 |
| Bond angles (°) | 0.840 | 0.480 | 0.680 | 0.880 | 0.750 |

Each dataset was collected from a single crystal. Values in parentheses are for highest-resolution shell.

**Extended Data Table 2 | Cryo-EM data collection and refinement statistics**

|  | CLN4-20 / cCpE / COP-2 / Nb (PDB: 9BEI , EMDB: 44479) |
|---|---|
| **Data collection and processing** | |
| Magnification | 120,000 |
| Voltage (kV) | 200 |
| Electron exposure (e–/Å$^2$) | 49.4 |
| Defocus range (μm) | -0.4-2.0 |
| Pixel size (Å) | 0.884 |
| Symmetry imposed | C1 |
| Initial particle images (no.) | 1,848,208 |
| Final particle images (no.) | 21,296 |
| Map resolution (Å) | 4.16 |
| FSC threshold | 0.143 |
| Map resolution range (Å) | 3.7 - 62.7 |
| | |
| **Refinement** | |
| Initial model used (PDB code) | 7DTM |
| Model resolution (Å) | 4.6 |
| FSC threshold | 0.5 |
| Model resolution range (Å) | 3.9 - 4.6 |
| Map sharpening $B$ factor (Å$^2$) | -74.2 |
| Model composition | |
| Non-hydrogen atoms | 6652 |
| Protein residues | 862 |
| Ligands | - |
| $B$ factors (Å$^2$) | |
| Protein | 321.9 |
| Ligand | - |
| R.m.s. deviations | |
| Bond lengths (Å) | 0.003 |
| Bond angles (°) | 0.781 |
| Validation | |
| MolProbity score | 2.35 |
| Clashscore | 21.04 |
| Poor rotamers (%) | 0.00 |
| Ramachandran plot | |
| Favored (%) | 90.73 |
| Allowed (%) | 9.15 |
| Disallowed (%) | 0.12 |

# Reporting Summary

## Statistics

For all statistical analyses, confirm that the following items are present in the figure legend, table legend, main text, or Methods section.

| n/a | Confirmed | |
|---|---|---|
| ☐ | ☒ | The exact sample size (*n*) for each experimental group/condition, given as a discrete number and unit of measurement |
| ☐ | ☒ | A statement on whether measurements were taken from distinct samples or whether the same sample was measured repeatedly |
| ☒ | ☐ | The statistical test(s) used AND whether they are one- or two-sided<br>*Only common tests should be described solely by name; describe more complex techniques in the Methods section.* |
| ☒ | ☐ | A description of all covariates tested |
| ☒ | ☐ | A description of any assumptions or corrections, such as tests of normality and adjustment for multiple comparisons |
| ☐ | ☒ | A full description of the statistical parameters including central tendency (e.g. means) or other basic estimates (e.g. regression coefficient) AND variation (e.g. standard deviation) or associated estimates of uncertainty (e.g. confidence intervals) |
| ☒ | ☐ | For null hypothesis testing, the test statistic (e.g. *F*, *t*, *r*) with confidence intervals, effect sizes, degrees of freedom and *P* value noted<br>*Give P values as exact values whenever suitable.* |
| ☒ | ☐ | For Bayesian analysis, information on the choice of priors and Markov chain Monte Carlo settings |
| ☒ | ☐ | For hierarchical and complex designs, identification of the appropriate level for tests and full reporting of outcomes |
| ☒ | ☐ | Estimates of effect sizes (e.g. Cohen's *d*, Pearson's *r*), indicating how they were calculated |

*Our web collection on statistics for biologists contains articles on many of the points above.*

## Software and code

Policy information about availability of computer code

| | |
|---|---|
| Data collection | Chromeleon (ThermoFischer Sci, v7.2.10) and Astra (Wyatt Tech., v8.0.2.5) for collecting SEC-MALS data |
| Data analysis | Coot (v0.9.5) for structure building; cryoSPARC v4.4.1 was used for cryoEM data processing; UCSF ChimeraX v1.4 and PyMOL (Schrödinger, v2.5.2) for structure visualization; XDS v Jan 10, 2022 (BUILT=20220220) and autoPROC (GlobalPhasing, version 20230222) for crystallographic data processing; Phenix (v1.20.1-4487) for molecular replacement phasing and structure refinement; MolProbity (v4.5.1) for structure validation; Biacore Insight Evaluation Software (Cytiva, v4.0.8.19878) for evaluating surface plasmon resonance measurements; Af2seq code is available at https://github.com/bene837/af2seq. ProteinMPNN, along with soluble trained weights is available at https://github.com/dauparas/ProteinMPNN. |

For manuscripts utilizing custom algorithms or software that are central to the research but not yet described in published literature, software must be made available to editors and reviewers. We strongly encourage code deposition in a community repository (e.g. GitHub). See the Nature Portfolio guidelines for submitting code & software for further information.

## Data

Policy information about availability of data

All manuscripts must include a data availability statement. This statement should provide the following information, where applicable:

- Accession codes, unique identifiers, or web links for publicly available datasets
- A description of any restrictions on data availability
- For clinical datasets or third party data, please ensure that the statement adheres to our policy

> All data are available in the main text or as supplementary materials. Atomic coordinates and structure factors of the reported X-ray structures have been deposited in the Protein Data Bank under accession numbers 8OYS (TBF_24), 8OYV (CLF_4), 8OYW (RPF_9), 8OYX (GLF_18), and 8OYY (GLF_32). CryoEM model has been deposited in the Protein Data Bank under accession number 9BEI and in the Electron Microscopy Data Bank under entry number 44479. PDB model 7TDM was used for rigid body docking into cryoEM density.

## Research involving human participants, their data, or biological material

Policy information about studies with human participants or human data. See also policy information about sex, gender (identity/presentation), and sexual orientation and race, ethnicity and racism.

| | |
|---|---|
| Reporting on sex and gender | Not applicable to the current study. |
| Reporting on race, ethnicity, or other socially relevant groupings | Not applicable to the current study. |
| Population characteristics | Not applicable to the current study. |
| Recruitment | Not applicable to the current study. |
| Ethics oversight | Not applicable to the current study. |

Note that full information on the approval of the study protocol must also be provided in the manuscript.

# Field-specific reporting

Please select the one below that is the best fit for your research. If you are not sure, read the appropriate sections before making your selection.

☒ Life sciences ☐ Behavioural & social sciences ☐ Ecological, evolutionary & environmental sciences

For a reference copy of the document with all sections, see nature.com/documents/nr-reporting-summary-flat.pdf

# Life sciences study design

All studies must disclose on these points even when the disclosure is negative.

| | |
|---|---|
| Sample size | For protein design expression and characterization, 20 designs were tested for IGF, 18 for BBF, 25 for TBF, 57 for GLF, 13 for CLF, 15 for RPF, 16 for GGC, 7 for CLN1, 6 for CLN4, 15 for iGLF, and 15 for aGLF. Designs were chosen according to top scoring in silico prediction metrics. |
| Data exclusions | Particles were excluded during 2D and 3D classification during cryoEM reconstruction. Removal of suboptimal particles is standard practice in single-particle cryoEM and is necessary to obtain homogeneous reconstructions. |
| Replication | Solubility expression experiments were performed a single time due to the robustness of expression conditions. SPR binding experiments were measured a single time with multiple concentrations and a negative control in parallel on a separate channel, to rule out unspecific binding events. |
| Randomization | Randomization is not applicable to this study as no live animals or human subjects were involved. |
| Blinding | Analyses in this manuscript were not blinded, as no live animals or human subjects were involved. Blinding is not standard practice for the presented in vitro experiments. In silico analyses were automated, no user intervention could introduce bias. |

# Reporting for specific materials, systems and methods

We require information from authors about some types of materials, experimental systems and methods used in many studies. Here, indicate whether each material, system or method listed is relevant to your study. If you are not sure if a list item applies to your research, read the appropriate section before selecting a response.

## Materials & experimental systems

| n/a | Involved in the study |
|-----|----------------------|
| ☒ | Antibodies |
| ☒ | Eukaryotic cell lines |
| ☒ | Palaeontology and archaeology |
| ☒ | Animals and other organisms |
| ☒ | Clinical data |
| ☒ | Dual use research of concern |
| ☒ | Plants |

## Methods

| n/a | Involved in the study |
|-----|----------------------|
| ☒ | ChIP-seq |
| ☒ | Flow cytometry |
| ☒ | MRI-based neuroimaging |

