## [Peer Review File · Nature]

Manuscript Title: Computational design of soluble and functional membrane protein analogues

Reviewer Comments & Author Rebuttals

Reviewer Reports on the Initial Version:

Referees' comments:

Referee #1 :

The authors present a computational pipeline that allows for the design of complex folds such as IgG's, beta barrels, and TIM barrels, which are challenging to design computationally. They use a variation of a previously established pipeline that integrates AlphaFold2, Rosetta, and ProteinMPNN. This method is applied to "challenging folds", such as the IgGs, TIM barrel, and beta barrel.

The main application of this work is to design folds found in membrane proteins but not in soluble proteins. Membrane proteins often exhibit unique topologies not found in soluble forms. The motivation for the work thus is to see whether the limited overlap in topology is observed in water soluble versus membrane proteins can be attributed to an intrinsic property of membrane proteins from, or rather simply be a consequence of the environment.

The authors find that the sequence optimizing component, ProteinMPNN, needs to be trained exclusively on soluble proteins in order to yield soluble, folded proteins. The designed sequences have minimal sequence homology with native ones yet mostly reproduce the folds attempted, albeit with backbone RMSD ranging from 2 Å to 5.4 Å. Interestingly, the sequences obtained recovered only some of the evolutionary conserved signatures – for example, most of the G in TIM barrel loops, but not the proline-rich sequence in the transmembrane domains; other features not recovered may have functional roles.

Although the concept of solubilizing membrane proteins to elucidate details on their structure is not new and has been explored in previous studies, this paper contains a very powerful computational pipeline that should be generalizable. In general, the initial motivation for these efforts was to provide structural information on membrane protein structure, when structural studies were challenging (e.g. very early designs on heme-binding proteins inspired by Cyt bc complex, or ion channels). Structures of these early designs are available (PNAS, 2004, DeGrado and colleagues). However, this motivation has become somewhat less relevant due to technical improvements in structure determination of membrane proteins. The authors cite most of the existing literature, with the exception of work by Shuguang Zhang using a simplified QTY code, which has proven surprisingly simple and successful (reviewed in Protein Design: From the Aspect of Water Solubility and Stability, Chem. Rev. 2022, 122:14085–14179, <https://doi.org/10.1021/acs.chemrev.1c00757>). No structures are available, however, for QTY designs. One should note that GPCRs have intrinsic dynamics that underpin their activity in signaling, thus replicating a "fixed" state per se is not biologically or functionally relevant--other than the challenge of achieving a fold not found in water soluble proteosome; that bridge, however, was crossed long ago with the design of Top7.

As it is often the case, the de novo designed proteins are very stable to thermal denaturation. The biophysical characterization presented, is limited to temperature challenge, whereas chemical denaturation would be more informative of the interactions that stabilize the protein.

Although this is undoubtedly an advance in the application of LLMs to protein design, neither the level of novelty or the application raises to the general interest for a Nature paper.

Referee #2:

The manuscript from Goverde et al. is an important contribution to the fields of de novo protein design, protein folding and applications of deep learning. The authors demonstrate that novel sequences can be generated that adopt target folds of complex protein topologies, including membrane proteins. To my knowledge, this is the first demonstration of deep learning algorithms to successfully achieve complex folds, with the notable addition that soluble analogues of integral membrane proteins could be designed.

This work combines two previously published methods, AF2seq and ProteinMPNN into “AF2seq-MPNN” and as such, could be viewed as an iterative advance. However, I believe this manuscript is an important addition to field. Interest to the broader readership of Nature would be increased by better describing potential applications of this technology in particular with regards to generating soluble membrane proteins.

Major points

- In terms of methodology, the primary advance is combining AF2seq and ProteinMPNN. Comparing Figs S1 and S2 demonstrates why this was done, since AF2seq-generated structures passed to ProteinMPNN succeed in generating final target fold structures for both the TIM-barrel and GPCR cases, whereas backbone structures from x-ray, backrub and MD simulation succeed only in the TIM-barrel case and fail completely on the GPCR test case.

The reasons as to why the other structure methods fail is not investigated, and this work would benefit from additional analysis. Was the GPCR structure (6FFI) in the ProteinMPNN training set? In the ProteinMPNN work (Dauparas et al., Science, 2022), it is noted that “Training with backbone noise improves model performance for protein design”. If the final ProteinMPNN model was trained with backbone noise, does the AF2seq process essentially impart the correct amount of backbone noise? Although a few methods were assessed (backrub and MD simulation) might there be a simpler way to add noise to the input structures?

It would be interesting to see how the input backbone structures compare, for example by calculating all backbone dihedral angles and creating a UMAP plot, to see where the structure-generating methods cluster (x-ray, backrub, MD, AF2seq).

- Along similar lines, Fig 1d: why do the AF2seq backbones generate so many more diverse

sequences? Is this because the input structures are themselves more diverse (greater dynamic range in Figs S1a, S2a)? The dihedral UMAP plot may give an indication of structural diversity.

- The authors note “AF2-based design approaches have been shown to generate plausible protein backbones, however, their performance in sequence design was suboptimal as evidenced by the low experimental success rates”. Although it is difficult to quantify “success”, since it is dependent on the number of generated sequences that are taken for experimental validation and the criteria of success, the broad success rate in the manuscript under review appears to be in-line with the papers cited:

Designs taken for expression Passing SEC-MALS % success

Wicky, Science, 2022 [HALs] 96 21 22

Goverde et al, Protein Sci, 2023 39 7 17.9

IGF 19 4 21

BBF 25 6 24

TIM TBF 25 5 20

CLF 13 3 23

RPF 15 3 20

GLF 56 10 17.8

- The authors note sequences are generated “without the need for experimental optimization or empirical restraints”. However, the manuscript documents an empirical optimization of the computational procedure, by first applying MPNNbias and then training and applying MPNNsol to generate membrane protein folds with suitable hydrophilic residues. It should therefore be noted that future similar applications of this procedure may require additional empirical optimization of the computational procedure.

- Please clarify this sentence: “the GPCR X-ray target contained an endolysin domain that was inserted into one of the loops to enhance protein solubility for crystallization (residues 679 to 838). Therefore, we predicted the GPCR sequence using the X-ray target as a template” – does this mean the GPCR sequences that were generated also contained an endolysin-like domain? If so, we do not know if the GLF designs are soluble due to the AF2seq-MPNNsol method or if they are soluble due to the endolysin-like domain. From the sequence length of the GLF designs, this does not appear to be the case (there is no endolysin-like domain), but the manuscript would benefit from making this clear to the reader.

- Potential applications are referred to but are somewhat over-stated and rather general: “Another exciting perspective is the creation of soluble analogues of membrane proteins that retain many of the native features of the original membrane proteins, such as enzymatic or transport functions, which could greatly accelerate the study of their function in more biochemically accessible soluble formats” – is a valid statement; however since their natural function is clearly related to the position and relation to the membrane these aspects may be compromised. Further “Similarly, this would be critical to facilitate the development of novel drugs and therapies that target this challenging class of proteins” – this is overstated and so should be toned down or further guidance on how this technology would be enabling for the development of novel drugs should be given.

A significant disadvantage of the GPCR sequences is that the GLF generated do not contain the conserved toggle switches thought to be crucial for the key conformational changes that relate to activation upon agonist binding (such as DRY motif, NPxxY motif). The current protocol would therefore require significant work to create 'functionally active', ligand-binding, soluble GPCRs that can undergo relevant conformational changes. It should be noted that creation of soluble, functionally-active GPCRs would require further optimization using the AF2seq-MPNNsol method.

- The generation of soluble membrane proteins is an intriguing development, but it should be noted in the text that these have been engineered in the past using physics-based methods; for example, see "Computational design of water-soluble analogues of the potassium channel KcsA, Slovic et al., PNAS, 2004".

Minor points

- It would be beneficial to the reader to have an idea of the similarity between the AF2seq-generated sequences and those generated by ProteinMPNN.
- It is interesting that in the membrane protein section, the AF2seq sequences had a low fraction of apolar surface residues yet did not express soluble constructs. What is the distribution across the CLF/RPF/GLF targets?
- Fig. S18b caption: "sequence diversity of all BBF designs" I believe should read "sequence diversity of all GLF designs".
- Ref 71: the correct citation for the AMBER ff99SB used in AF2 is "Hornak et al., Proteins, 2006".

Reviewers: Callum Dickson and Fiona Marshall, Novartis

Referee #4:

Summary of the key results:

This article describes a workflow for designing protein sequences adopting desired folds by leveraging recent advancements in AF2 (sequence-to-structure prediction) and ProteinMPNN (structure-to-sequence prediction). At each iteration, a structure is predicted for the sequence, and a loss derived from the similarity and confidence of AF2 output to the target is computed. The structure is deconstructed to a sequence via ProteinMPNN, and by continuously iterating, a sequence predicted to fold into the target fold is produced. Authors demonstrate that on backbone fold design, the method outperforms pre-AF2 methods. The authors then demonstrate the precision of such an approach via experimental characterization, and examine the well-posed problem of designing soluble proteins adopting folds primarily observed in membrane-bound proteins. Using ML-proposed designs and with various rounds of filtering, authors successfully produce soluble analogs of membrane proteins with interesting biophysical characterization and crystallization to

enable new directions for membrane proteomic research.

Originality and significance:

The problem-finding in this work is done well from a machine learning (ML) perspective. Recent works have pointed towards the eminently-realizable potential of being able to design sequences that adopt particular folds with help from deep learning, but few have applied it to wet lab verified experiments. This work further finds a subproblem that is well-suited to this advance, and yields interesting novel insights into the membrane proteome. This elucidation of the biophysics of membrane proteins can inform many other aspects of membrane proteome research, e.g. its evolution and bioengineering use.

While the method is sound, the novelty might be lacking for the Nature readership. From the perspective of pushing the ML for proteins community forward, it would be more exciting to see methods that suggests a fundamental paradigm shift in how we conceive the problem. Using AlphaFold2 for design is a paradigm that the community has been exploring since its release, and computationally-enabled with the open-source ColabDesign repository (<https://github.com/sokrypton/ColabDesign>). Verkuil et al (<https://www.biorxiv.org/content/10.1101/2022.12.21.521521v1>) and Hie et al. (<https://www.biorxiv.org/content/10.1101/2022.12.21.521526v1>) uses similar paradigms involving iterative tinkering of the sequence to optimize against AlphaFold2 produced outputs. To further enforce this workflow as a fundamental paradigm that the community should embrace, the work may benefit from addressing other paradigms for sequence design to form specific folds: Retraining language models to make them structure aware (e.g. <https://arxiv.org/abs/2302.01649>) Simultaneous generation of sequence and structure conditioned on fold specification (<https://www.biorxiv.org/content/10.1101/2023.05.08.539766v1>). Recent diffusion based models such as RFDiffusion (<https://www.nature.com/articles/s41586-023-06415-8>)

Data & methodology: validity of approach, quality of data, quality of presentation

From the perspective of pushing forward the machine learning for protein design community:

- Direct comparisons to recent deep learning baselines (e.g. those mentioned in the section above) and ablations would certainly be helpful. Though the shortcoming is not critical, given that the primary contribution of the work is to membrane proteomics, the strength of the proposed method currently feels under-supported.
- Are model weights publicly available? <https://github.com/bene837/af2seq/tree/main> points to placeholder directories. Would be great to clarify where they are.
- Using AF2 in single sequence mode begs the question of if using OmegaFold (<https://www.biorxiv.org/content/10.1101/2022.07.21.500999v1>) or ESMFold (<https://www.biorxiv.org/content/10.1101/2022.07.20.500902v1>) can work better, since they

demonstrate better results in the single sequence regime.

- Chaining these models together greatly amplifies hallucination effects and model uncertainty. It is known that AF2 can be subjected to adversarial attacks (<https://arxiv.org/abs/2305.08929>) and produce confident predictions for sequence which do not fold experimentally (<https://www.biorxiv.org/content/10.1101/2023.05.23.541774v1>). Tying back to the overall comment of proposing a paradigm shift, I would be more convinced if this was addressed.

Conclusions: robustness, validity, reliability:

As addressed previously, the work provides a compelling investigation and harness recent methodological progress to an interesting problem. The strength of this workflow paradigm over other approaches is less robust.

Suggested improvements: experiments, data for possible revision

The article may benefit from addressing a few points:

- What fraction of designs passed the selection threshold? I.e. if one were to sample more proteins, what fraction might be expected to be viable?
- More information on how the soluble variant of ProteinMPNN is trained would be helpful, as it seems to be a key ingredient in the empirical success. For example, knowing the dataset size can inform how portable this retraining technique is to other domains.
- How exactly were the sequences initialized? Lines 462-465 gives a rough outline of this, but it is unclear what the motivation for this was. The authors point towards a lack of structural diversity, and it would be interesting to see how different initializations affect the diversity-quality trade off.
- How "confident" were the designs? A figure demonstrating how the argmax value in the PSSM changes overtime would be interesting.
- What were the weights for the individual loss terms for the final model? This is interesting both from a reproducibility perspective, as well as deriving insight into how the frame-based loss, distogram loss, and pLDDT / pTM interplay.

References: appropriate credit to previous work?

These works may be pertinent for the references list:

- Validation of de novo designed water-soluble and transmembrane proteins by in silico folding and melting. Hermosilla et al., 2023 (<https://www.biorxiv.org/content/10.1101/2023.06.06.543955v2>)
- Structure-informed Language Models Are Protein Designers. Zheng et al., 2023 (<https://arxiv.org/abs/2302.01649>)

Clarity and context: lucidity of abstract/summary, appropriateness of abstract, introduction and conclusions

- The work introduces the method with AF2seq; while this is great for understanding the thought evolution behind the project, the article becomes more difficult to follow without referencing that work. It may be better to describe the method as is, and appending notes on which components came from AF2-design.
- Caption of Fig 3.: The terms "design" and "target" may be confusing for general audiences, and it would be clearer to specify which are crystallized structures from the PDB and which are AF2 predicted, etc.
- The machine learning methods would greatly benefit from an expanded write-up, such as including pseudocode for key algorithms. Previously published ML for protein design publications in this journal (e.g. Jumper et al., Watson et al.) includes supplemental material at a level of detail that makes it possible to essentially implement the code from scratch; the write-up in its current form falls short of this.

Minor:

- Caption of Supp. Fig 1 should be "designed", not "designed"
- Line 192: open bracket

Overall Recommendation:

The work combines progress in fold-specific sequence design with deep learning to open problems in understanding transmembrane proteins. This sets a great example of problem-finding for future ML for protein design works. My background does not allow me to meaningfully comment on the impact that this work can have on transmembrane protein research; for the ML component alone, I might expect an algorithm in Nature to propose a fundamental paradigm shift, which the work falls short of in its current form. The work is strong, and would be very interesting to readers in more targeted publications than Nature. I would overall recommend Reject or Accept with Major Revisions, depending on the strength of the experimental contribution as gauged by other reviewers.

Author Rebuttals to Initial Comments:

Referees' comments – Goverde & Pacesa, et al. 2023

Referee #1 :

The authors present a computational pipeline that allows for the design of complex folds such as IgG's, beta barrels, and TIM barrels, which are challenging to design computationally. They use a variation of a previously established pipeline that integrates AlphaFold2, Rosetta, and ProteinMPNN. This method is applied to “challenging folds”, such as the IgGs, TIM barrel, and beta barrel.

The main application of this work is to design folds found in membrane proteins but not in soluble proteins. Membrane proteins often exhibit unique topologies not found in soluble forms. The motivation for the work thus is to see whether the limited overlap in topology is observed in water soluble versus membrane proteins can be attributed to an intrinsic property of membrane proteins from, or rather simply be a consequence of the environment.

The authors find that the sequence optimizing component, ProteinMPNN, needs to be trained exclusively on soluble proteins in order to yield soluble, folded proteins. The designed sequences have minimal sequence homology with native ones yet mostly reproduce the folds attempted, albeit with backbone RMSD ranging from 2 Å to 5.4 Å. Interestingly, the sequences obtained recovered only some of the evolutionary conserved signatures – for example, most of the G in TIM barrel loops, but not the proline-rich sequence in the transmembrane domains; other features not recovered may have functional roles.

Although the concept of solubilizing membrane proteins to elucidate details on their structure is not new and has been explored in previous studies, this paper contains a very powerful computational pipeline that should be generalizable. In general, the initial motivation for these efforts was to provide structural information on membrane protein structure, when structural studies were challenging (e.g. very early designs on heme-binding proteins inspired by Cyt bc complex, or ion channels). Structures of these early designs are available (PNAS, 2004, DeGrado and colleagues). However, this motivation has become somewhat less relevant due to technical improvements in structure determination of membrane proteins. The authors cite most of the existing literature, with the exception of work by Shuguang Zhang using a simplified QTY code, which has proven surprisingly simple and successful (reviewed in Protein Design: From the Aspect of Water Solubility and Stability, Chem. Rev. 2022, 122:14085–14179, <https://doi.org/10.1021/acs.chemrev.1c00757>). No structures are available, however, for QTY designs. One should note that GPCRs have intrinsic dynamics that underpin their activity in signaling, thus replicating a “fixed” state per se is not biologically or functionally relevant--other than the challenge of achieving a fold not found in water soluble proteosome; that bridge, however, was crossed long ago with the design of Top7.

R: We thank the reviewer for the assessment. Indeed, many of the typical sequence signatures of some of these folds were not recapitulated in our designs, however, this was intended, as it demonstrates that the sequence space of these folds is far greater than that found in nature (most likely due to functional constraints). Although impressive and important work has been done previously by other groups to generate soluble membrane proteins, these methods were usually applied to a very specific subgroup of membrane proteins, and do not offer a

generalisable approach. We believe that our approach would offer an exciting alternative, with deep learning approaches providing greater accuracy and success rates than before.

While the QTY code methodology provides similar advantages, such as its simplicity and generalisability, GPCRs solubilised using this method still require either mild detergent (Tegler et al 2020 Sci. Reports) or purification under denaturing conditions and refolding in a strong reducing environment to be extracted (Zhang et al 2018 PNAS; Qing et al 2019 PNAS; Hao et al 2020 QRB Discovery). Our designs can be easily expressed in *E.coli* with no special buffer requirements or refolding, and result in high protein yields.

This is most likely a specific problem of GPCR folds, where the presence of a hydrophilic core and then the introduction of a hydrophilic surface would cause problems during protein folding in solution. Therefore, utilising our methodology to generate stable hydrophobic cores for these folds circumvents this problem, at the cost of certain functional aspects, such as conformational changes. Nevertheless, functional interacting epitopes important for G-protein, drug, or antibody binding can be preserved, as we demonstrate in the newly added data shown in Figures 5 and 6.

Lastly, only AlphaFold2-based structural models of QTY designs have been described so far (Skuhersky et al 2021 Life; Smorodina et al 2022 Sci. Reports). The structures closely match the native GPCRs, but examination of the methodology shows that these mutated/designed proteins were predicted using multiple-sequence alignments (MSAs). This has been in the past shown to make AF2 insensitive to mutations (Buel & Walters 2022 NSMB; Roney & Ovchinnikov 2022 Phys. Rev. Lett; Pak et al 2023 PLoS ONE; Stein & Mchaourab 2023 bioRxiv; <https://alphafold.ebi.ac.uk/faq>), therefore most plausible mutants would be predicted “correctly”. This is in contrast to our single-sequence prediction, where no MSAs were utilised, therefore AF2 has to rely on learned structural principles. In addition, our experimental structures demonstrate that soluble versions of membrane proteins do recapitulate the native folds faithfully. Relative to the point brought up about the design of Top7, where a new complete fold was created, we see this as a different challenge than that of what we sought to ask in this current work, which was directed to topologies existent in membrane environments.

Overall, our approach provides several advantages over existing techniques when it comes to design of complex folds with functional sites, especially when accuracy for conformation-specific design is required. We demonstrate this in both GPCR folds and claudin folds, where functional interaction sites can be either preserved during design or grafted into protein scaffolds which benefit from the built-in stability and overall biochemical tractability.

As it is often the case, the de novo designed proteins are very stable to thermal denaturation. The biophysical characterization presented, is limited to temperature challenge, whereas chemical denaturation would be more informative of the interactions that stabilize the protein.

R: We agree that chemical denaturation would provide us with deeper insights into the nature of the design’s stability, unfortunately, we were not able to perform these experiments due to technical limitations of our CD setup. Given the wealth of structural and functional information included in the manuscript we deemed that there is a good level of understanding of the interactions occurring within the structures of these designs. From the structures we can observe that likely the high thermal stability of our designs stems from the tightly packed and stable hydrophobic cores that were designed by AF2seq+MPNN, and which are often

observed in de novo designed proteins (Wicky et al 2022 Science, Dauparas et al 2022 Science, Watson et al 2023 Nature, and others).

Although this is undoubtedly an advance in the application of LLMs to protein design, neither the level of novelty nor the application raises to the general interest for a Nature paper.

R: We would like to highlight that large language models (LLMs) were not used in the design of proteins in this study, all protein design procedures were solely structure-based. In more technical terms, at the time of submission there were several manuscript utilizing similar AlphaFold2-based methods for design (Jendrusch et al 2021 bioRxiv, Moffat et al 2021 bioRxiv, Wicky et al 2022 Science), however, these previously published methods apply stochastic Markov Chain Monte Carlo sampling to apply mutations randomly and then predict these sequences with AlphaFold2 and score, which can be computationally expensive. Our approach backpropagates through the AlphaFold2 network and directly utilises the trained network weights to generate plausible protein sequences compatible with the provided fold. This approach is similar to the approach developed by Wang et al., however, for their approach there was no structural validation of the AF2-generated designs (Wang et al. 2022). We acknowledge that while our approach is not entirely novel, the complexity of the folds designed, their functionalization, as well as the generation of conformation-specific designs are all significant advances that stand in the cutting edge of the field of protein design and such approaches will be broadly applicable in areas of fundamental biology and biotechnology. Moreover, our computational design efforts are presented with a significant amount of experimental characterization and structure determination data that is uncommon in many computational protein design papers.

Referee #2 :

The manuscript from Goverde et al. is an important contribution to the fields of de novo protein design, protein folding and applications of deep learning. The authors demonstrate that novel sequences can be generated that adopt target folds of complex protein topologies, including membrane proteins. To my knowledge, this is the first demonstration of deep learning algorithms to successfully achieve complex folds, with the notable addition that soluble analogues of integral membrane proteins could be designed.

This work combines two previously published methods, AF2seq and ProteinMPNN into "AF2seq-MPNN" and as such, could be viewed as an iterative advance. However, I believe this manuscript is an important addition to field. Interest to the broader readership of Nature would be increased by better describing potential applications of this technology in particular with regards to generating soluble membrane proteins.

R: We thank the reviewers for their kind words.

Major points

In terms of methodology, the primary advance is combining AF2seq and ProteinMPNN. Comparing Figs S1 and S2 demonstrates why this was done, since AF2seq-generated structures passed to ProteinMPNN succeed in generating final target fold structures for both

the TIM-barrel and GPCR cases, whereas backbone structures from x-ray, backrub and MD simulation succeed only in the TIM-barrel case and fail completely on the GPCR test case.

The reasons as to why the other structure methods fail is not investigated, and this work would benefit from additional analysis. Was the GPCR structure (6FFI) in the ProteinMPNN training set? In the ProteinMPNN work (Dauparas et al., Science, 2022), it is noted that “Training with backbone noise improves model performance for protein design”. If the final ProteinMPNN model was trained with backbone noise, does the AF2seq process essentially impart the correct amount of backbone noise? Although a few methods were assessed (backrub and MD simulation) might there be a simpler way to add noise to the input structures?

R: We have checked and the 6FFI structure (along with many other GPCR structures) was indeed present in the original ProteinMPNN training set, which would explain why designing using original weights results in the design of membrane sequences. The training of soluble MPNN was performed the same way as the original MPNN, including backbone noise, with the exception of excluding membrane structures. We performed a comprehensive analysis of various levels of backbone noise applied to the crystal structure template of the TIM barrel fold during sequence design and included these results as a new Supplementary Figure S2.

To summarise, levels of 0.5 Å noise are comparable to the crystal structure only. Noise levels above 0.7 Å mostly result in low confidence designs and adversarial sequences. While levels of 0.6 Å show good RMSD values compared to the starting template and good confidence values in single sequence predictions, they display higher levels of sequence recovery and e-values (when BLASTed against a database of natural proteins) compared to AF2seq. We also note that such analysis would look very different for other folds, especially more complex natural folds like GPCR, which cannot be predicted in single sequence mode without prior AF2seq optimisation.

It would be interesting to see how the input backbone structures compare, for example by calculating all backbone dihedral angles and creating a UMAP plot, to see where the structure-generating methods cluster (x-ray, backrub, MD, AF2seq).

R: We have generated a UMAP plot of dihedral angles of (a) input structures generated using different backbone perturbation methods and of (b) resulting MPNN redesigned and repredicted output designs, including crystal structures where different levels of Gaussian noise (based on comment above) were applied to the backbone during inference. It is noteworthy that MD-based backbones cluster far away from other backbone generation methods (both input and output), most likely due to large variation of backbone conformations, as apparent from large RMSD values (Supplementary Figure 1a). While Rosetta Backrub methods and gaussian noise perturbations occupy a similar UMAP region in the angles of output MPNN designs (b), AF2seq backbones are located in a separate and very tight cluster in the output but cluster around the crystal structure and backrub methods in the input (a). While difficult to interpret, we hypothesize that templates resulting from AF2seq, similarly to AF2 predictions, sample more frequently native-like configurations in contrast to physics-based methods such as Rosetta or MD.

Along similar lines, Fig 1d: why do the AF2seq backbones generate so many more diverse sequences? Is this because the input structures are themselves more diverse (greater dynamic range in Figs S1a, S2a)? The dihedral UMAP plot may give an indication of structural diversity.

R: The sequence diversity of resulting MPNN designs when using AF2seq backbones as input stems primarily from the completely novel protein cores that are designed during the AF2seq step. To illustrate this point we updated Fig 1c to include conservation of the core residues and surface residues. ProteinMPNN generally exhibits high levels of sequence conservation in the core (Supplementary Figure 2, and Dauparas et al. 2022 Science), and by generating novel core sequences prior to MPNN optimisation we can achieve much higher levels of sequence novelty compared to other approaches.

The authors note “AF2-based design approaches have been shown to generate plausible protein backbones, however, their performance in sequence design was suboptimal as evidenced by the low experimental success rates”. Although it is difficult to quantify “success”, since it is dependent on the number of generated sequences that are taken for experimental validation and the criteria of success, the broad success rate in the manuscript under review appears to be in-line with the papers cited:

Designs taken for expression Passing SEC-MALS % success

Wicky, Science, 2022 [HALs] 96 21 22

Goverde et al, Protein Sci, 2023 39 7 17.9

IGF 19 4 21

BBF 25 6 24

TIM TBF 25 5 20

CLF 13 3 23

RPF 15 3 20

GLF 56 10 17.8

R: We would like to point out that we tested the indicated amounts of proteins for expression but for the membrane analogues actually only few were further purified and taken for SEC-MALS analysis. In addition, many of the expressed constructs were taken directly from the AF2seq pipeline, which later on became clear to result in non-purifiable proteins, to give a

contrast to our MPNN-optimised approach. Here is an updated table with a breakdown of the designs that perhaps illustrates this better, with success rates based on the designs with biophysical characterisation:

Fold	Tested AF2seq designs	Tested AF2seq+MPNN designs	Soluble AF2seq designs	Soluble success %	Further purified	Passing SEC-MALS	CD+SEC-MALS success %
IGF	10	9	7	37	7	4	57
BBF	8	17	17	68	11	6	54
TIM	12	13	14	56	9	5	55
CLF	10	27	13	35	5	4	80
RPF	7	20	17	63	5	5	100
GLF	12	37	33	85	10	10	100

The authors note sequences are generated “without the need for experimental optimization or empirical restraints”. However, the manuscript documents an empirical optimization of the computational procedure, by first applying MPNNbias and then training and applying MPNNsol to generate membrane protein folds with suitable hydrophilic residues. It should therefore be noted that future similar applications of this procedure may require additional empirical optimization of the computational procedure.

R: We apologise for the ambiguity in our statement, we were referring to the parametric design of certain folds, such as the TIM barrel in Huang et al. 2016, where extensive design constraints were applied to achieve such fold, followed by visual inspection and rational optimization of the structure. In all our design trajectories, only the structure of the desired fold is needed to guide design. The creation of MPNNsol was then a specific adaptation that was necessary to address the unique challenge of designing soluble membrane protein folds. Traditional ProteinMPNN was recognizing the membrane folds and designing them as such, although the original weights still remain effective for the design of soluble proteins. Gathering from the results of our work, MPNNsol can likely be applied to the solubilization of other membrane folds without additional tuning.

Please clarify this sentence: “the GPCR X-ray target contained an endolysin domain that was inserted into one of the loops to enhance protein solubility for crystallization (residues 679 to 838). Therefore, we predicted the GPCR sequence using the X-ray target as a template” – does this mean the GPCR sequences that were generated also contained an endolysin-like domain? If so, we do not know if the GLF designs are soluble due to the AF2seq-MPNNsol method or if they are soluble due to the endolysin-like domain. From the sequence length of the GLF designs, this does not appear to be the case (there is no endolysin-like domain), but the manuscript would benefit from making this clear to the reader.

R: Thank you for the suggestion, we have further clarified this point in the methods section. We removed the endolysin domain prior to design and replaced it with a generic linker. Therefore, all GLF designs consist of purely the GPCR fold and are soluble on their own.

Potential applications are referred to but are somewhat over-stated and rather general: “Another exciting perspective is the creation of soluble analogues of membrane proteins that retain many of the native features of the original membrane proteins, such as enzymatic or transport functions, which could greatly accelerate the study of their function in more biochemically accessible soluble formats” – is a valid statement; however since their natural function is clearly related to the position and relation to the membrane these aspects may be compromised. Further “Similarly, this would be critical to facilitate the development of novel drugs and therapies that target this challenging class of proteins” – this is overstated and so should be toned down or further guidance on how this technology would be enabling for the development of novel drugs should be given.

R: We thank the reviewers for their input. We have revised our language to better reflect the possibilities that our soluble analogues provide. Additionally, we have now included proof of principle experiments demonstrating that the functional epitopes of natural membrane proteins can be recapitulated with our soluble analogues using two complementary procedures (Figure 5 & 6).

Firstly, functional epitopes can simply be grafted onto the designed soluble analogues, as demonstrated by chimeras of the Ghrelin receptor which could still be recognised by an antibody raised against the natural membrane protein.

Secondly, we have extended our design methodology to constrain the natural functional epitopes during design. This way, we could generate soluble versions of claudins that are bound by the *Clostridium perfringens* enterotoxin - a common pathogenic agent in gastrointestinal disease. More importantly, we were able to use this method to perform conformation-specific design of GPCR proteins, in both active and inactive state, to facilitate or preclude G-protein binding, as demonstrated by our experimental binding assays.

We believe that these experiments underscore the potential of our approach to investigate functional aspects of membrane proteins in solution and thereby accelerate the search for protein-based and, potentially, small molecule-based therapeutics targeting these folds.

A significant disadvantage of the GPCR sequences is that the GLF generated do not contain the conserved toggle switches thought to be crucial for the key conformational changes that relate to activation upon agonist binding (such as DRY motif, NPxxY motif). The current protocol would therefore require significant work to create ‘functionally active’, ligand-binding, soluble GPCRs that can undergo relevant conformational changes. It should be noted that creation of soluble, functionally-active GPCRs would require further optimization using the AF2seq-MPNNsol method.

R: We acknowledge that this might have been an overstatement on our part. While the fully *de novo* designed soluble analogues would not be able to recapitulate this characteristic, we believe that our novel approach, outlined in our response above, where functionally relevant epitopes are constrained during design, might lead to partial or in some cases full restoration of such function. However, further research would be necessary to demonstrate this. In the meantime, we have appropriately rephrased this section of the manuscript.

The generation of soluble membrane proteins is an intriguing development, but it should be noted in the text that these have been engineered in the past using physics-based methods;

for example, see “Computational design of water-soluble analogues of the potassium channel KcsA, Slovic et al., PNAS, 2004”.

R: Thank you, we have added references to rational- (QTY code) and force field-based methods to the main text.

Minor points

It would be beneficial to the reader to have an idea of the similarity between the AF2seq-generated sequences and those generated by ProteinMPNN.

R: We have generated a plot showcasing the TIM barrel fold (PDB: 5BVL) sequence conservation between: 1) MPNN designed sequences and the input PDB
2) AF2seq designed sequences and the input PDB
3) AF2seq designed sequences vs AF2seq-MPNN designed sequences

We observe similar trends as previously, where MPNN recovers 40-50% of the input sequence, confirming that the sequence novelty originates from AF2seq design.

It is interesting that in the membrane protein section, the AF2seq sequences had a low fraction of apolar surface residues yet did not express soluble constructs. What is the distribution across the CLF/RPF/GLF targets?

Initially, we did not test many AF2seq designs for other folds, as we observed very early on that such designs rarely result in soluble expression. We have now tested an appropriate number of high-ranking AF2seq designs for both CLF and RPF analogues, and observe very poor solubility of the designs despite their low surface hydrophobicity.

Fig. S18b caption: “sequence diversity of all BBF designs” I believe should read “sequence diversity of all GLF designs”.

R: Thank you, this has been corrected.

Ref 71: the correct citation for the AMBER ff99SB used in AF2 is “Hornak et al., Proteins, 2006”.

R: Thank you for spotting this error, we have corrected the citation.

Reviewers: Callum Dickson and Fiona Marshall, Novartis

Referee #4 :

Summary of the key results:

This article describes a workflow for designing protein sequences adopting desired folds by leveraging recent advancements in AF2 (sequence-to-structure prediction) and ProteinMPNN (structure-to-sequence prediction). At each iteration, a structure is predicted for the sequence, and a loss derived from the similarity and confidence of AF2 output to the target is computed. The structure is deconstructed to a sequence via ProteinMPNN, and by continuously iterating, a sequence predicted to fold into the target fold is produced. Authors demonstrate that on backbone fold design, the method outperforms pre-AF2 methods. The authors then demonstrate the precision of such an approach via experimental characterization, and examine the well-posed problem of designing soluble proteins adopting folds primarily observed in membrane-bound proteins. Using ML-proposed designs and with various rounds of filtering, authors successfully produce soluble analogs of membrane proteins with interesting biophysical characterization and crystallization to enable new directions for membrane proteomic research.

Originality and significance:

The problem-finding in this work is done well from a machine learning (ML) perspective. Recent works have pointed towards the eminently-realizable potential of being able to design sequences that adopt particular folds with help from deep learning, but few have applied it to wet lab verified experiments. This work further finds a subproblem that is well-suited to this advance, and yields interesting novel insights into the membrane proteome. This elucidation of the biophysics of membrane proteins can inform many other aspects of membrane proteome research, e.g. its evolution and bioengineering use.

R: We thank the reviewer for the kind words.

While the method is sound, the novelty might be lacking for the Nature readership. From the perspective of pushing the ML for proteins community forward, it would be more exciting to

see methods that suggests a fundamental paradigm shift in how we conceive the problem. Using AlphaFold2 for design is a paradigm that the community has been exploring since its release, and computationally-enabled with the open-source ColabDesign repository (<https://github.com/sokrypton/ColabDesign>). Verkuil et al (<https://www.biorxiv.org/content/10.1101/2022.12.21.521521v1>) and Hie et al. (<https://www.biorxiv.org/content/10.1101/2022.12.21.521526v1>) uses similar paradigms involving iterative tinkering of the sequence to optimize against AlphaFold2 produced outputs.

To further enforce this workflow as a fundamental paradigm that the community should embrace, the work may benefit from addressing other paradigms for sequence design to form specific folds:

Retraining language models to make them structure aware (e.g. <https://arxiv.org/abs/2302.01649>)

Simultaneous generation of sequence and structure conditioned on fold specification (<https://www.biorxiv.org/content/10.1101/2023.05.08.539766v1>).

Recent diffusion based models such as RFDiffusion (<https://www.nature.com/articles/s41586-023-06415-8>)

R: We acknowledge the reviewer's concern regarding the novelty of our approach. There have been other studies utilising AlphaFold2 for design (Jendrusch et al 2021 bioRxiv, Moffat et al 2021 bioRxiv, Wicky et al 2022 Science), however, in most cases AlphaFold2 is used to score the designed sequences, rather than design them directly. AF2seq functions similarly to ColabDesign (developed independently in parallel), where we backpropagate through the AlphaFold2 network to generate plausible sequences for the target fold, and therefore utilise its learned knowledge of protein structure for design. However, as stated in our replies to other reviewers, we believe that the strength of our approach is its generalisability and high experimental success rate in producing native-like topologies, focusing especially on producing soluble versions of membrane proteins, which would be difficult to achieve with language model-based approaches, as these have the tendency to generate significantly more false positive designs than AF2 (Hermosilla et al 2023 bioRxiv). We do acknowledge that structure conditioned RF diffusion in combination with soluble MPNN could also work in generating soluble membrane topologies. The main usage of AF2seq is the generation of realistic backbones with plausible starting sequences, which is in contrast to RFDiffusion, where no starting sequence is provided. Furthermore, in the latest version of our manuscript we have added significant additional *in silico* and experimental data. This elevates the work beyond the design of structural folds and introduces the design of function into this very important class of proteins in biology, bringing another dimension to this work that surpasses and strengthens the computational aspect.

Data & methodology: validity of approach, quality of data, quality of presentation

From the perspective of pushing forward the machine learning for protein design community: Direct comparisons to recent deep learning baselines (e.g. those mentioned in the section above) and ablations would certainly be helpful. Though the shortcoming is not critical, given that the primary contribution of the work is to membrane proteomics, the strength of the proposed method currently feels under-supported.

R: We thank the reviewer for the suggestion. While such an analysis would be very insightful, we believe the strength of our approach lies in its generalisability and the biochemical problem it was applied to. On the computational side we presented multiple metrics for the dependency of backbone generation and sequence design steps. We also studied the effect of different levels of noise during the inference of MPNN-based sequence design, and lastly the levels of sequence diversity in the different regions of the protein. While these are not the more traditional ablation tests, which in some of the modules we are using are complicated to perform without retraining (e.g. AF module), in our opinion we have investigated our design approach to a significant level of depth. In addition, the large body of experimental data, specifically structural data, should attest and support the robustness of the approach in a level that computational benchmarks do not provide.

Are model weights publicly available? <https://github.com/bene837/af2seq/tree/main> points to placeholder directories. Would be great to clarify where they are.

R: We use the original AlphaFold2 monomer model weights for AF2seq design. We have now clarified this in the methods and data availability sections and added instructions to the github repository.

Using AF2 in single sequence mode begs the question of if using OmegaFold (<https://www.biorxiv.org/content/10.1101/2022.07.21.500999v1>) or ESMFold (<https://www.biorxiv.org/content/10.1101/2022.07.20.500902v1>) can work better, since they demonstrate better results in the single sequence regime.

R: Although language-model based prediction algorithms have been previously suggested to be applicable to the structure prediction of designed proteins, it has since been shown that they may be suboptimal predictors of de novo proteins (<https://www.biorxiv.org/content/10.1101/2023.06.06.543955v2>). In such pipelines, language model embeddings serve as replacements for multiple-sequence alignments to provide co-evolutionary information and guide structure prediction. In the case of designed proteins, this may prove misleading, and therefore result in confident predictions of even spurious protein sequences. Single sequence mode in AF2 can overcome this obstacle by not relying on any co-evolutionary information, but rather on the learned structural properties of proteins. Although false positives may still occur, in our experience, high confidence scores resulting from single sequence AF2 predictions of MPNN-optimised designs seem to be very good indicators of experimental success.

Chaining these models together greatly amplifies hallucination effects and model uncertainty. It is known that AF2 can be subjected to adversarial attacks (<https://arxiv.org/abs/2305.08929>) and produce confident predictions for sequence which do not fold experimentally (<https://www.biorxiv.org/content/10.1101/2023.05.23.541774v1>). Tying back to the overall comment of proposing a paradigm shift, I would be more convinced if this was addressed.

R: While hallucination effects are certainly a great concern, especially in protein design, we in fact observe that including ProteinMPNN in the design pipeline actually rescues any adversarial sequences that AF2seq might produce. This is supported by experimental data in Fig 4e, where af2seq designs rarely, if ever, can be produced experimentally, while their MPNN-optimised counterparts work most of the time.

Conclusions: robustness, validity, reliability:

As addressed previously, the work provides a compelling investigation and harness recent methodological progress to an interesting problem. The strength of this workflow paradigm over other approaches is less robust.

Suggested improvements: experiments, data for possible revision
The article may benefit from addressing a few points:

- What fraction of designs passed the selection threshold? I.e. if one were to sample more proteins, what fraction might be expected to be viable?

R: To provide a clearer idea of the designs that pass the filters, we refer to Supp Fig. S5, which shows that in silico success rates are dependent on the size and complexity of the fold. We acknowledge that this doesn't give a good representation of what designs pass all of the filters, hence we summarise all designs in the table below which has also been added to the supplementary information as table S2.

Fold	total designs	designs passing filter	in silico success (%)
IGF	150	34	23%
BBF	72	26	36%
TBF	144	84	58%
CLF	750	52	7%
RPF	1769	32	2%
GLF	1063	176	17%

From our previous experience, we have found that TM-score and pLDDT of the AF2 predicted model correlate with the in vitro success rate (Goverde et al. 2022). A correlation between the RMSD and pLDDT of the AF2 generated model and the in vitro success rate was also previously observed (Hermosilla et al., 2023, Bryant et al., 2022). Hence, we expect a reduction in experimental success rates if lower the pLDDT threshold to increase in silico success rates. We could potentially relax the TM-score threshold, which would increase in

silico success rates, but result in more structural diversity between the designs and the target fold.

More information on how the soluble variant of ProteinMPNN is trained would be helpful, as it seems to be a key ingredient in the empirical success. For example, knowing the dataset size can inform how portable this retraining technique is to other domains.

R: For more details regarding the training of proteinMPNN we refer to the original paper Dauparas et al 2022 Science. The training of soluble proteinMPNN is identical to the original proteinMPNN, with the exception that membrane proteins are completely excluded from the dataset. We have found that 6FFI (GPCR), amongst other membrane targets, was indeed part of the training set, which would explain why the original proteinMPNN was redesigning these proteins as membrane proteins. The set of excluded membrane proteins can be found in the ProteinMPNN [github repo](https://github.com/dauparas/ProteinMPNN/blob/main/soluble_model_weights/excluded_PDBs.csv) https://github.com/dauparas/ProteinMPNN/blob/main/soluble_model_weights/excluded_PDBs.csv

How exactly were the sequences initialized? Lines 462-465 gives a rough outline of this, but it is unclear what the motivation for this was. The authors point towards a lack of structural diversity, and it would be interesting to see how different initializations affect the diversity-quality trade off.

R: We thank the reviewer for raising this point. This analysis was performed in our previous study (Goverde et al. 2022). In the starting sequence, the amino acid identities are assigned according to which Secondary Structural Element (SSE) they are expected to form according to DSSP. Next, helical residues get assigned alanines, beta-sheets valines, and loops glycines (see Fig S2. in Goverde et al 2022). In the current study, several initialization strategies were tried out, such as polyalanine and random sequences. However, even though these sequences would eventually converge, for quick convergence (within 500 iterations) SSE-initialization was needed (See Figure 1c in Goverde et al. 2022). Since AF2 is deterministic in single sequence prediction mode, we mutate 10% of the sequence to get different starting points resulting in different points of convergence. This results in generated sequences with only 10-30% sequence similarity (Goverde et al. 2022 Figure S3). Since these sequences exhibited low e-values (e-value < 0.05), i.e. de novo sequences, we did not experiment with other initialization methods.

How "confident" were the designs? A figure demonstrating how the argmax value in the PSSM changes overtime would be interesting.

When we plot the overall loss and RMSD to the template design over design iterations, we observe that it converges relatively quickly, around 200 iterations, depending on design fold (as observed previously in Goverde et al 2022). Trends in the design of soluble proteins (panel a and b) and soluble membrane analogues (panels c and d) are similar. The confidence intervals are calculated based on averaged losses/RMSDs of all five AF2 models for all design trajectories for the indicated folds.

What were the weights for the individual loss terms for the final model? This is interesting both from a reproducibility perspective, as well as deriving insight into how the frame-based loss, distogram loss, and pLDDT / pTM interplay.

R: The weights used for design of all folds were $W_{\text{fape}} = 1.0$, $W_{\text{pLDDT}} = 0.2$ and $W_{\text{pTM}} = 0.2$. If the trajectories didn't result in significant convergence, we found that adding a distogram loss (W_{dist}) of 0.5 helped with convergence. The sequences that converged were then used as 'soft starts' to seed new design trajectories. Here we found that the distogram loss wasn't necessary for convergence anymore, and hence it was disabled as FAPE is the higher resolution structural loss.

As a note, if we set the W_{pLDDT} and W_{pTM} loss weights too high we found that there was a strong preference to generate helices without matching the target template. This can be explained by the helix being the most local secondary structural element which is easiest to predict.

References: appropriate credit to previous work?

These works may be pertinent for the references list:

- Validation of de novo designed water-soluble and transmembrane proteins by in silico folding and melting. Hermosilla et al., 2023 (<https://www.biorxiv.org/content/10.1101/2023.06.06.543955v2>)
- Structure-informed Language Models Are Protein Designers. Zheng et al., 2023 (<https://arxiv.org/abs/2302.01649>)

R: We thank the reviewer for their suggestions. The first publication is indeed relevant and we have included it in our main text, however, we would like to mention that this preprint was not published at the time of submission. The second publication provides an interesting

alternative to our purely-structure based approach to design, but in our opinion is a more distant family of methods that is perhaps more out of the scope of our work.

Clarity and context: lucidity of abstract/summary, appropriateness of abstract, introduction and conclusions

The work introduces the method with AF2seq; while this is great for understanding the thought evolution behind the project, the article becomes more difficult to follow without referencing that work. It may be better to describe the method as is, and appending notes on which components came from AF2-design.

R: We thank the reviewer for the suggestion. We have now adjusted the text and figures to indicate where sequences were generated with only AF2seq and which ones were MPNN optimised following AF2seq (AF2seq-MPNN).

Caption of Fig 3.: The terms "design" and "target" may be confusing for general audiences, and it would be clearer to specify which are crystallized structures from the PDB and which are AF2 predicted, etc.

R: We thank the reviewer for the suggestion, the labels were indeed confusing, we have now appropriately labelled all structures in the figures to indicate what structure is shown and how it was obtained in parentheses, for example TBF_24 (x-ray) or Design (AF2).

The machine learning methods would greatly benefit from an expanded write-up, such as including pseudocode for key algorithms. Previously published ML for protein design publications in this journal (e.g. Jumper et al., Watson et al.) includes supplemental material at a level of detail that makes it possible to essentially implement the code from scratch; the write-up in its current form falls short of this.

R: We thank the reviewer for this suggestion. As both AF2seq (Goverde et al 2022) and ProteinMPNN (Dauparas et al 2022) have been previously described in detail in their respective publications, we wanted to dedicate more focus to the applications of these pipelines when combined.

Minor:

- Caption of Supp. Fig 1 should be "designed", not "designed"
- Line 192: open bracket

R: Thank you, we corrected these errors.

Overall Recommendation:

The work combines progress in fold-specific sequence design with deep learning to open problems in understanding transmembrane proteins. This sets a great example of problem-finding for future ML for protein design works. My background does not allow me to meaningfully comment on the impact that this work can have on transmembrane protein research; for the ML component alone, I might expect an algorithm in Nature to propose a fundamental paradigm shift, which the work falls short of in its current form. The work is strong, and would be very interesting to readers in more targeted publications than Nature. I would overall recommend Reject or Accept with Major Revisions, depending on the strength of the experimental contribution as gauged by other reviewers.

R: We thank the reviewer for the assessment. As we stated in our response to other reviewers, we agree that similar computational approaches have been described previously. However, the novelty of our work lies in the refined and generalisable procedure to generate: complex natural-like folds with high experimental success, proteins with embedded functional sites and conformation-specific states of computationally designed proteins. Additionally, our application of this approach to the solubilisation and functionalization of membrane proteins is likely to be of general interest to the audience of Nature and will find many interesting applications in the screening of drugs and inhibitors of therapeutically relevant targets.

Reviewer Reports on the First Revision:

Referees' comments:

Referee #1:

The authors revised the paper extensively following reviewers' concerns. The manuscript is streamlined and improved in clarity. Overall, the experimental characterization is thorough.

Referee #2:

We thank Goverde & Pacesa et al. for their detailed response to our comments and the revised manuscript. We find the responses suitably address our concerns. Further, Goverde & Pacesa et al. have performed significant additional work to generate and characterize functionally active soluble membrane proteins, which addresses our key criticism of the initial work. As such, we believe this manuscript will be of high value to Nature readers.

We identified a small number of minor points:

- Fig. 1 caption: The “c,” is missing in the caption (“sequence diversity. , Novelty of generated sequences”)
- Fig. 6d: Kd measurements are provided for ICL3-specific antibody binding to the GLF-Ghrelin chimeras. For reference, is the Kd for the ICL3-specific antibody and wild-type Ghrelin receptor known?
- Design of active/inactive A2AR states: The PDB structures are referenced but it might benefit the reader to explicitly include the PDB codes for the active/inactive templates, as has been done for the earlier protein design sections.

This review was conducted by Fiona Marshall with input from Callum Dickson (Novartis Biomedical Research).

Referee #2 (Remarks on code availability):

The AF2seq github appears well documented. However, the paper under review details a pipeline combining AF2seq, ProteinMPNN and AF. Is any code made available to run the full pipeline (and therefore for others to reproduce this work)?

We have not installed or run any examples.

Referee #4:

Thanks to the authors for providing updated commentary and manuscript. The clarifications in the Methods and Materials on loss weighting and initialization were helpful for substantiating the methodological thrust of the paper. The table of design fractions that succeed (provided in the commentary for the reviewer) was also very useful; it would be nice to see this table in the supplemental/methods along with comparisons to other methods.

As the authors note in their response, the main merits of the paper is not its methodological innovation, but its contribution to understanding soluble proteomes. With the updated information on the ML methods, I think that the method is sensible on the whole. It remains limited in its innovation, and to some extent, scientific rigor, i.e. it fails to rigorously elucidate which parts of the pipeline were most important. My concerns with the main ML method has been mentioned in my previous review, and since the core method was not updated, I will not repeat them here.

On the whole, I find the experimental verifications regarding solubility and structural fidelity to membrane analogs to be impressive, and agree that many in the Nature viewership may find this interesting. However, I think this primary impact can be highlighted more clearly. As the authors note in their comments to the reviewer, the approach "will find many interesting applications in the screening of drugs and inhibitors of therapeutically relevant targets", but this important justification is not highlighted in the text or results. One naive question to ask, for example, is how membrane proteins existing in solution can be used for developing therapeutics, since this new placement removes its transport capabilities across the cell membrane. Clarifying exactly downstream works can build upon the ability to generate soluble membrane protein analogs would justify why this work is of Nature-level impact.

On the whole, I'd be more comfortable recommending a straightforward "Accept" if there were experiments (or at least text) that connects soluble membrane protein analogs to therapeutically-adjacent applications. As it currently stands, the focus seems to still be on the ML pipeline, and that authors could harness it to design proteins that befit their aims. I unfortunately think that this contribution is not substantial or novel enough for the Nature publication. I recommend a weak accept for this revised version, because results are indeed convincing that authors can generate proteins with impressive solubility and fidelity to the target membrane folds, and it is a fascinating challenge to how to define and understand the soluble proteome. However, this primary impact of the paper is weakly highlighted, and lots of room exist for strengthening how downstream works can build upon the findings regarding solubility. My present opinion remains that future works can glean little with respect to how this pipeline can be used for non-solubility associated design efforts.

Referee #4 (Remarks on code availability):

Yes, I was able to install and run the main script using instructions provided.

Author Rebuttals to First Revision:

Referees' comments – Goverde & Pacesa & Goldbach, et al. 2024

Referee #1 :

The authors revised the paper extensively following reviewers' concerns. The manuscript is streamlined and improved in clarity. Overall, the experimental characterization is thorough.

R: We are very grateful to the reviewer for their kind feedback.

Referee #2 :

We thank Goverde & Pacesa et al. for their detailed response to our comments and the revised manuscript. We find the responses suitably address our concerns. Further, Goverde & Pacesa et al. have performed significant additional work to generate and characterize functionally active soluble membrane proteins, which addresses our key criticism of the initial work. As such, we believe this manuscript will be of high value to Nature readers.

R: We thank the reviewers for their positive assessment and are glad we have addressed their concerns.

We identified a small number of minor points:

- Fig. 1 caption: The “c,” is missing in the caption (“sequence diversity. , Novelty of generated sequences”)

R: Thank you, we have added the missing caption.

- Fig. 6d: Kd measurements are provided for ICL3-specific antibody binding to the GLF-Ghrelin chimeras. For reference, is the Kd for the ICL3-specific antibody and wild-type Ghrelin receptor known?

R: Unfortunately, no affinity measurement of the antibody binding to the membrane receptor were performed in the original publication.

- Design of active/inactive A2AR states: The PDB structures are referenced but it might benefit the reader to explicitly include the PDB codes for the active/inactive templates, as has been done for the earlier protein design sections.

R: We have added the PDB codes to the figure legend in addition to the methods.

This review was conducted by Fiona Marshall with input from Callum Dickson (Novartis Biomedical Research).

The AF2seq github appears well documented. However, the paper under review details a pipeline combining AF2seq, ProteinMPNN and AF. Is any code made available to run the full pipeline (and therefore for others to reproduce this work)?

We have not installed or run any examples.

R: We usually run each part of the pipeline separately, so we have uploaded the notebooks used to run Af2seq, SolubleMPNN and AF2 to the github.

Referee #4 :

Thanks to the authors for providing updated commentary and manuscript. The clarifications in the Methods and Materials on loss weighting and initialization were helpful for substantiating the methodological thrust of the paper. The table of design fractions that succeed (provided in the commentary for the reviewer) was also very useful; it would be nice to see this table in the supplemental/methods along with comparisons to other methods.

R: Thank you for your feedback, we have included the table as part of the manuscript.

As the authors note in their response, the main merits of the paper is not its methodological innovation, but its contribution to understanding soluble proteomes. With the updated information on the ML methods, I think that the method is sensible on the whole. It remains limited in its innovation, and to some extent, scientific rigor, i.e. it fails to rigorously elucidate which parts of the pipeline were most important. My concerns with the main ML method has been mentioned in my previous review, and since the core method was not updated, I will not repeat them here.

On the whole, I find the experimental verifications regarding solubility and structural fidelity to membrane analogs to be impressive, and agree that many in the Nature viewership may find this interesting. However, I think this primary impact can be highlighted more clearly. As the authors note in their comments to the reviewer, the approach "will find many interesting applications in the screening of drugs and inhibitors of therapeutically relevant targets", but this important justification is not highlighted in the text or results. One naive question to ask, for example, is how membrane proteins existing in solution can be used for developing therapeutics, since this new placement removes its transport capabilities across the cell membrane. Clarifying exactly downstream works can build upon the ability to generate soluble membrane protein analogs would justify why this work is of Nature-level impact.

On the whole, I'd be more comfortable recommending a straightforward "Accept" if there were experiments (or at least text) that connects soluble membrane protein analogs to therapeutically-adjacent applications. As it currently stands, the focus seems to still be on the ML pipeline, and that authors could harness it to design proteins that benefit their aims. I unfortunately think that this contribution is not substantial or novel enough for the Nature publication. I recommend a weak accept for this revised version, because results are indeed convincing that authors can generate proteins with impressive solubility and fidelity to the target membrane folds, and it is a fascinating challenge to how to define and understand the soluble proteome. However, this primary impact of the paper is weakly highlighted, and lots of room exist for strengthening how downstream works can build upon the findings regarding solubility. My present opinion remains that future works can glean little with respect to how this pipeline can be used for non-solubility associated design efforts.

R: Thank you for the reviewer's suggestion on providing further context regarding the impact of our study. We agree that the applications of soluble membrane analogues are the most exciting aspect of this work and we have outlined several of them in the latter part of the discussion. This includes the study of certain enzymatic functions present in the membrane or, most commonly, the screening of agonists and antagonists of cell surface receptors, such as GPCRs, which have potential to become novel therapeutics. However, we have yet to demonstrate small molecule binding activity in the soluble space with such fold.

Referee #4 (Remarks on code availability):

Yes, I was able to install and run the main script using instructions provided.